# Self-Predictive Representations for Combinatorial Generalization in Behavioral Cloning

**Daniel Lawson**[1,2,*]   **Adriana Hugessen**[1,2,*]   **Charlotte Cloutier**[1]
**Glen Berseth**[1,2,†]   **Khimya Khetarpal**[1,3,†]
[1]Mila   [2]Université de Montréal   [3]Google DeepMind

## Abstract

While goal-conditioned behavior cloning (GCBC) methods can perform well on in-distribution training tasks, they do not necessarily generalize zero-shot to tasks that require conditioning on novel state-goal pairs, i.e. *combinatorial generalization*. In part, this limitation can be attributed to a lack of temporal consistency in the state representation learned by BC; if temporally correlated states are properly encoded to similar latent representations, then the out-of-distribution gap for novel state-goal pairs would be reduced. We formalize this notion by demonstrating how encouraging long-range temporal consistency via successor representations (SR) can facilitate generalization. We then propose a simple yet effective representation learning objective, BYOL-$\gamma$ for GCBC, which theoretically approximates the successor representation in the finite MDP case through self-predictive representations, and achieves competitive empirical performance across a suite of challenging tasks requiring combinatorial generalization.

## 1 Introduction

Generalization has been a long-standing goal in machine learning and robotics. Recently, large-scale supervised models for language and vision have demonstrated impressive generalization when trained over vast amounts of data. In robotics, this has motivated large-scale behavior cloning (BC) models trained on offline datasets of diverse demonstrations (Ghosh et al., 2024; Kim et al., 2024). However, these models still suffer from a lack of generalization. In particular, while BC methods can perform well on tasks directly observed in the dataset, they often fail to perform zero-shot transfer to tasks requiring novel combinations of in-distribution behavior, known as *combinatorial generalization*. In the robotics domain, where demonstration data is time-intensive and costly to produce, simply scaling the dataset is often not possible. Hence, achieving this type of generalization algorithmically will be critical to unlocking the potential for large-scale supervised policy training.

The property of combinatorial generalization has been previously formalized as the ability to "stitch" (Ghugare et al., 2024). Here, stitching refers to the ability of a policy to reach a goal state from a start state when trained on a dataset, which provides sufficient coverage of the path to the goal, but which does not contain a single complete trajectory of the path. The lack of stitching observed in goal-conditioned behavioral cloning (GCBC) and, more generally, supervised learning, can be understood through the inductive biases of the model. By construction, BC methods do not encode the inductive bias that the observed data are generated from a Markov decision process (MDP). In contrast, reinforcement learning (RL) policies that are trained via temporal difference (TD) learning directly utilize the structure of the MDP to pass information through time using dynamic programming. Offline RL (Levine et al., 2020) has been proposed as a method for achieving stitching in policies trained on offline datasets. However, these methods are challenging to scale due to the instability of bootstrapping in TD learning when combined with fully offline training. Scaling has been more successful with supervised methods, such as in robotics, where training robot foundation models with BC (Ghosh et al., 2024; Kim et al., 2024) on large-scale datasets (O'Neill et al., 2024; Khazatsky et al., 2024) can lead to more general-purpose policies.

---

\* Co first author. † Equal advising. Project page: `https://self-pred-bc.github.io/`.
Correspondence: `{daniel.lawson, adriana.knatchbull-hugessen}@mila.quebec`

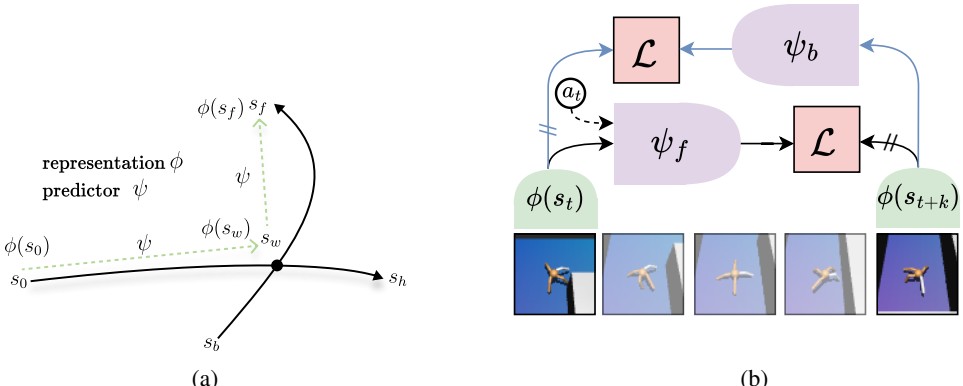

(a)                                                                    (b)

Figure 1: **(a) Self-predictive Representations.** We consider training on trajectories like, $s_0 \to s_h$ and $s_b \to s_f$, which intersect at $w$, and then evaluating on a task like $s_0 \to s_f$, requiring combinatorial generalization. **(b) Representation learning with BYOL-$\gamma$.** We predict future state representations $\phi(s_{t+k})$ via $\psi_f(\phi(s_t), a)$, and also predict backwards with $\psi_b(\phi(s_{t+k}))$. The target offset is sampled geometrically: $k \sim \text{geom}(1 - \gamma)$. Stop-gradients are denoted by //. We provide more details on the training procedure $\mathcal{L}$ in Section 4.2.

A goal-conditioned policy being general-purpose implies it has learned an implicit world model of the environment (Richens et al., 2025). From this intuition, a key desiderata is to make a policy's representations align with the (latent) dynamics of the underlying environment, in order to obtain a more robust goal-conditioned policy. However, an open question here is which representation learning objective best achieves this property. We begin to investigate this question with *Bootstrap Your Own Latent* (BYOL) framework (Grill et al., 2020), which in RL, learns a representation space through predicting future latent states (Schwarzer et al., 2020), without requiring negative samples nor TD-learning. While the standard BYOL objective has been shown to learn representations capturing spectral information about the one-step transition dynamics (Khetarpal et al., 2025), we find that a key property is to capture temporally extended information, leading us to (1) propose a novel objective, **BYOL-$\gamma$** which predicts future states geometrically (Figure 1b), and (2) present a unifying framework (Table 1) for understanding objectives related to the successor representation (Blier et al., 2021), including contrastive learning, BYOL, BYOL-$\gamma$, and a novel application of a known TD-based approximation of the SR as an auxiliary loss for BC. Namely, we quantity how these methods uniquely behave when applied to data collected by a mixture of policies which is encountered in practical BC settings.

In the finite, single-policy MDP case, we show that BYOL-$\gamma$ approximates the successor representation. As with other non-TD methods, in the mixture-policy case, BYOL-$\gamma$ corresponds to approximating a mixture of SRs, however, with less pessimism than existing contrastive objectives. Qualitatively, the BYOL-$\gamma$ objective learns representations that encode long-range temporal distance between states on mixture datasets more faithfully, as compared to TD, than contrastive learning (Figure 2). Empirically, on the challenging OGBench suite (Park et al., 2025), we demonstrate that BYOL-$\gamma$ augmented GCBC outperforms all other methods (Table 2), on average, and is robust to combinatorial generalization with increasing horizons (Figure 3). Our representation can also be extended to hierarchical setups (Appendix C), which leads to further improvements in generalization.

## 2 RELATED WORK

**Stitching in Supervised Methods.** Outcome (goals or return)-conditioned behavioral cloning (OCBC) methods (Schmidhuber, 2020; Chen et al., 2021; Emmons et al., 2022) provide a simple and scalable alternative to traditional offline RL (Levine et al., 2020) methods. However, these methods do not properly "stitch" and generalize to unseen outcomes (Brandfonbrener et al., 2022; Ghugare et al., 2024). To reduce this problem, various works have proposed augmenting training data used by BC methods. Some work incorporates methdology from offline RL to label returns or goals for downstream SL (Char et al., 2022; Yamagata et al., 2023). Other work has considered relabeling goals through clustering states (Ghugare et al., 2024), which relies on a good distance metric, or utilized planing Zhou et al. (2024) for goal relabeling, or generative models to synthesize new trajectories (Lu et al., 2023; Lee et al., 2024). Rather than using models to generate data, combinatorial generalization

can be achieved by planning with generative models (Luo et al., 2025). In this work, we neither require explicit Q-learning, generative models, or perform explicit planning.

**Representation learning in RL.** Our objective is most closely related to approaches using auxiliary **BYOL objectives in online RL** (Gelada et al., 2019; Schwarzer et al., 2020; Ni et al., 2024; Voelcker et al., 2024). These objectives can help with sample-efficiency, such as in challenging, partially observed environments with sparse rewards, or with noisy states. Additionally, self-predictive dynamics models are used in planning and model-based RL (François-Lavet et al., 2019; Ye et al., 2021; Hansen et al., 2022). Various works have also characterized the dynamics of BYOL objectives in the RL setting, showing that BYOL objectives capture spectral information about the policy's transitions (Tang et al., 2023; Khetarpal et al., 2025). In the offline setting, how well Joint Embedding Predictive Architecture (JEPA) world models generalize when used for explicit planning has been studied Sobal et al. (2025), however, not for combinatorial generalization. Additionally, certain representation structures for value functions, namely quasimetrics (Liu et al., 2023; Wang et al., 2023; Wang and Isola, 2022; Myers et al., 2024) can also lead to policies that better generalize to longer horizons (Myers et al., 2025a).

**Successor Representation** (SR) (Dayan, 1993) objectives, such as successor features (SF) (Barreto et al., 2017), and the successor measure (SM) (Blier et al., 2021) have been widely used for generalization and transfer in reinforcement learning (Carvalho et al., 2024). Similarly to BYOL, these objectives have been used for representation learning in RL (Lan et al., 2022; Farebrother et al., 2023). While prior BYOL methods either perform 1-step, or relatively short fixed n-step prediction, neither of these choices directly approximate the successor measure. Our setup is most related to temporal representation alignment (TRA) (Myers et al., 2025b), which recently proposed using contrastive learning as an auxiliary objective for BC to improve combinatorial generalization. In this work, we further build on the relationship between the SM and combinatorial generalization, and propose new objectives which can lead to better performance.

## 3 BACKGROUND

**Controlled Markov Process.** We consider goal-conditioned decision-making, with states $\mathcal{S}$, actions $\mathcal{A}$, goals $g \in S$, initial state distribution $p_0(s)$, dynamics $p(s_{t+1} \mid s_t, a)$, and with policies $\pi(a|s, g)$.

**Successor Representation (SR) and Successor Measure (SM).** In a finite MDP, the *successor representation* (SR) (Dayan, 1993) of a policy is: $M^\pi(s, s') := \mathbb{E}\left[\sum_{t \geq 0} \gamma^t \mathbb{1}_{(s_{t+1} = s')} \mid s_0 = s, \pi\right]$ We use the convention of counting from $s_{t+1}$, writing in matrix form $M^\pi = \sum_{t \geq 0} \gamma^t (P^\pi)^{t+1}$. The transition matrix for policy $\pi$ is $P^\pi$, with $P^\pi_{i,j} = \sum_a \pi(a|s = i) P_{i,a,j}$, where $P_{i,a,j} = p(s_{t+1} = j \mid s_t = i, a)$. The successor representation also satisfies the bellman equation, $M^\pi = P^\pi + \gamma P^\pi M^\pi = P^\pi (I - \gamma P^\pi)^{-1}$. For a fixed policy, the successor representation describes a type of temporal distance between states. The *successor measure* (SM) (Blier et al., 2021) extends SR to continuous spaces $S$: $M^\pi(s, X) := \sum_{t \geq 0} \gamma^t P(s_{t+1} \in X \mid s) \, \forall X \subset S$. We also define the *normalized* successor representation, or measure $\tilde{M}^\pi = (1 - \gamma) M^\pi$. In the finite case, the normalized successor representation $\tilde{M}^\pi$ has rows that sum to one like transitions $P^\pi$. We also define the state occupancy via $M^\pi(s') = \mathbb{E}_{s \sim p_0(s)}[M^\pi(s_0, s')]$. Another quantity, *successor features* (SF) (Barreto et al., 2017) are the expected discounted sum of future features $\phi(s) \in \mathbb{R}^d$: $\psi^\pi(s) = \mathbb{E}\left[\sum_{t \geq 0} \gamma^t \phi(s_{t+1}) \mid s_0 = s, \pi\right]$. We can relate SFs to the SM with $\psi^\pi(s) = \int_{s'} M^\pi(s, s')\phi(s')$. These quantities can also condition an action, e.g. $M^\pi(s, a, s')$.

### 3.1 REPRESENTATION LEARNING

We begin with two representation learning methods that approximate the density of the SM.

**Contrastive Learning.** Temporal contrastive learning used in MDPs (Eysenbach et al., 2022) is related to a Monte Carlo (MC) approximation of the (discounted) successor measure. This can be implemented with an InfoNCE (van den Oord et al., 2019) loss that maximizes the similarity of a positive pair consisting of a state $s_t$ and a future state from the same trajectory $s_+$, and minimizing

the similarity of negative pairs consisting of $s_t$ and random states $s_-$:

$$\min_{\phi,\psi} \mathbb{E}_{\substack{s_t \sim p(s) \\ k \sim \text{geom}(1-\gamma) \\ s_+ = s_{t+k}, s_-^{2:N} \sim p(s)}} \left[ -\log \frac{e^{f(\psi(s_t),\phi(s_+))}}{\sum_{i=2}^{N} e^{f(\psi(s_t),\phi(s_-^i))}} \right] \tag{1}$$

A common choice for the energy function $f$ is the inner product $f(\psi(s)\phi(s_+)) = \psi(s)^T\phi(s_+)$. A key aspect to note is that the positive sample $s_+$ comes from an MC sample from $s_+ \sim M^\pi(s_t, s_+)$. The optimal solution to (1) gives $\tilde{M}^\pi(s, s_+) \approx C \exp(\psi(s_t)^T\phi(s_+)) \cdot p(s_+)$.

**Temporal-Difference Approximation of SR (TD-SR)** We consider a Forward-Backward (Touati and Ollivier, 2021)-like loss that approximates the successor measure for a fixed policy $\pi$ using TD learning, discussed by Touati et al. (2023), which we call TD-SR.

$$\min_{\phi,\psi} \mathbb{E}_{\substack{s_t \sim p(s), s' \sim p(s) \\ s_{t+1} \sim p^\pi(s_{t+1}|s_t)}} \left[ (\psi(s_t)^T\phi(s') - \gamma\bar{\psi}(s_{t+1})^T\bar{\phi}(s'))^2 \right] - 2\mathbb{E}_{\substack{s_t \sim p(s) \\ s_{t+1} \sim p^\pi(s_{t+1}|s_t)}} \left[ \psi(s_t)^T\phi(s_{t+1}) \right] \tag{2}$$

TD-SR learns an approximation of the successor measure with factorization $M^\pi(s, s_+) \approx \psi(s_t)^T\phi(s_+) \cdot p(s_+)$ using TD learning. Given transitions $(s_t, s_{t+1})$ sampled by a policy $\pi$, the second term relates to fitting $M^\pi(s_t, s_{t+1})$. Given an independently sampled state $s'$, the first term bootstraps an estimate of $M^\pi(s_t, s')$ from $\bar{M}^\pi(s_{t+1}, s')$, where $\bar{\phi}, \bar{\psi}$ denote stop-gradient operations. In Appendix D, we further elaborate on the relationship between the TD-SR loss, and CL. Particularly, in the limit, an n-step version of TD-SR is related to CL.

**BYOL.** We now look at an objective that captures information about single-step transitions instead of the successor measure. In the context of RL, self-predictive models jointly learn a latent space and a dynamics model through predicting future latent representations. Self-predictive models rely on latent bootstrapped targets (BYOL) (Grill et al., 2020), avoiding reconstruction (generative models), or negative samples (contrastive learning). Self-predictive models are an instance of joint-embedding predictive architectures (JEPAs) (LeCun, 2022; Garrido et al., 2024).

Given an encoder which produces a representation $z_t = \phi(s_t)$, and dynamics $\psi(z_{t+1}|z_t)$ for a fixed policy $\pi$, we minimize the difference between our prediction and target representation in latent-space:

$$\min_{\phi,\psi} \mathbb{E}_{s_t \sim p(s), s_{t+1} \sim p^\pi(s_{t+1}|s_t),} \left[ f(\psi(\phi(s_t)), \bar{\phi}(s_{t+1})) \right] \tag{3}$$

Where $f$ measures the discrepancy between representations, such as the squared $l_2$ norm, and $\bar{\phi}$ refers to an EMA target, or stop-gradient. Variants of this BYOL objective have been widely used to learn state abstractions, and work as an auxiliary loss to value-function learning (Gelada et al., 2019; Schwarzer et al., 2020; Ni et al., 2024). In finite MDPs, this objective captures spectral information about one-step transitions $P^\pi$ (Tang et al., 2023; Khetarpal et al., 2025), discussed in Appendix E.1.

## 3.2 COMBINATORIAL GENERALIZATION FROM OFFLINE DATA

We now shift focus on how we can learn policies from offline data using behavioral cloning, and then introduce a combinatorial generalization gap that arises in this setting.

We consider a **dataset** $\mathcal{D} = \{(s_0^i, a_0^i, \cdots, s_T^i, a_T^i)\}_{i=1}^N$, composed of trajectories generated by a set of unknown policies $\{\beta_j(a|s)\}$. **Goal Conditioned Behavioral Cloning (GCBC)** trains a policy $\pi_\Theta$ with maximum likelihood to reproduce the behaviors from the dataset. After sampling a current state, a goal is sampled as a future state from the same trajectory:

$$\max_{\pi_\Theta} \mathcal{L}_{\text{BC}}(\pi_\Theta) = \max_\pi \mathbb{E}_{\substack{\beta_j \sim p(\beta_j), s \sim M^{\beta_j}(s) \\ a \sim \beta_j(a \mid s), s_+ \sim M^{\beta_j}(s, s_+)}} \left[ \log \pi_\Theta(a|s, g = s_+) \right] \tag{4}$$

**Generalization gap.** While this policy can perform well in-distribution, the behavior cloning policy struggles to generalize to reach goals from states that are not in matching training trajectories. We now review a more formal definition of this type of generalization gap.

We consider Lemma 3.1 from Ghugare et al. (2024), which says there exists a single Markovian policy $\beta(a|s)$ that has the same occupancy as the mixture of $j$ policies: $M^\beta(s) = \mathbb{E}_{p(\beta_j)} \left[ M^{\beta_j}(s) \right]$. This

policy also has construction: $\beta(a \mid s) := \sum_j \beta_j(a \mid s)p(\beta_j \mid s)$, where $p(\beta_j \mid s)$ is the distribution over policies in $s$ as reflected by the dataset.

Using the successor measure of the individual policies, and the mixture policy, we can quantify a gap between accomplishing out-of-distribution tasks versus in-distribution training tasks (Ghugare et al., 2024):

$$\underbrace{\mathbb{E}_{\substack{s_0 \sim M^\beta(s_0) \\ s_g \sim M^\beta(s_0, s_g)}} \left[ u^{\pi_\Theta}(s_0, s_g) \right]}_{\textbf{tasks requiring combinatorial generalization}} - \underbrace{\mathbb{E}_{\substack{\beta_j \sim p(\beta_j),\, s_0 \sim M^{\beta_j}(s_0) \\ s_g \sim M^{\beta_j}(s_0, s_g)}} \left[ u^{\pi_\Theta}(s_0, s_g) \right]}_{\textbf{in-distribution training tasks}} \tag{5}$$

Here, $u$ is a performance metric of the policy $\pi_\Theta$ such as the success rate to reach $s_g$ from $s_0$. As we perform well on in-distribution tasks due to a correspondence to Equation (4), the BC policy has no guarantees for the first term. This is because after sampling a state, the goal is sampled from the successor measure of the mixture policy.

## 4 USING REPRESENTATIONS FOR COMBINATORIAL GENERALIZATION

In this section, we aim to reduce the aforementioned generalization gap. We consider a policy trained with the BC objective $\pi_\Theta$ to be made more robust to the tasks requiring combinatorial generalization through representation learning. We begin with a setup similar to Equation (5), but with a shared initial state $s_0$ for both the in-distribution and out-of-distribution task. For the in-distribution task, we sample a goal as before, labeled as $s_w$. However, for the out-of-distribution task, we sample a goal $s_f$ to be a state that can be reached by the mixture policy $\beta$ after $s_w$. (6):

$$\mathbb{E}_{\substack{\beta_j \sim p(\beta_j), s_0 \sim M^{\beta_j}(s) \\ s_w \sim M^{\beta_j}(s_0, s_w)}} \left[ \underbrace{\mathbb{E}_{s_f \sim M^\beta(s_w, s_f)}\left[ u^{\pi_\Theta}(s_0, s_f)) \right]}_{\textbf{extended task requiring generalization}} - \underbrace{u^{\pi_\Theta}(s_0, s_w)}_{\textbf{in-distribution task}} \right] \tag{6}$$

$$= \mathbb{E}_{\substack{\beta_j \sim p(\beta_j), s_0 \sim M^{\beta_j}(s) \\ s_w \sim M^{\beta_j}(s_0, s_w)}} \left[ \underbrace{\mathbb{E}_{s_f \sim M^\beta(s_w, s_f)}\left[ u^{\pi_\Theta}(s_0, \phi(s_f)) \right]}_{\textbf{want invariance with respect to future goals through } \phi} - u^{\pi_\Theta}(s_0, \phi(s_w)) \right] \tag{7}$$

Then, in Equation (7) we add a goal representation $\phi$ that processes the goal before going to policy $\pi_\Theta$. Intuitively, a policy could achieve the out-of-distribution task by first going from $s_0$ to $s_w$ (in-distribution), and then completing the remaining task $s_w$ to $s_f$. In essence, we want that when conditioning on $\phi(s_f)$, the policy should first go to $s_w$, which can be achieved by learning $\phi$, where $\phi(s_w)$ is similar to $\phi(s_f)$ (Myers et al., 2025b). More formally, for $s_f \sim M^\beta(s_w, s_f)$ we want an invariance $\phi(s_f) \approx \phi(s_w)$. From this observation, we can understand obtaining a representation $\phi$ related to the successor measure of the mixture policy $\beta$ can be beneficial.

The BYOL framework would be a simple way to learn these representations that capture temporal dependencies. However, in Section 5 we demonstrate that a simple BYOL objective empirically leads to limited generalization when used as an auxiliary loss. Intuitively, the standard BYOL objective directly approximates the one-step transition dynamics, not the successor measure, and so struggles to capture relationships between distant states, separated by several trajectories.

### 4.1 BYOL-$\gamma$: CONNECTING SELF-PREDICTIVE OBJECTIVES TO THE SUCCESSOR REPRESENTATION

To build better self-predictive representations, we propose BYOL-$\gamma$ which allows us to use the BYOL framework to capture temporally extended information, i.e. *successor representations*. Given a state $s_t$, a BYOL objective samples prediction targets from one-step transition as in Equation (3). However, we make a modification to predict empirical samples from the normalized successor measure:

$$\mathcal{L}_{\text{BYOL-}\gamma}(\phi, \psi) = \mathbb{E}_{s_t \sim\ p(s), k \sim \text{geom}(1-\gamma), s_{t+k} \sim p^\pi(s_{t+k}|s_t)} \left[ f(\psi(\phi(s_t)), \bar{\phi}(s_{t+k})) \right] \tag{8}$$

Where $f$ refers to an energy function, $\phi$ refers to the encoder, and $\psi$ the predictor. With $\gamma = 0$, we have $s_{t+k} = s_{t+1}$ corresponding to an approximation of the one-step transitions, recovering the base

| Method | Approx. $\tau \sim \beta$ | Approx. $\tau \sim \{\beta_j\}$ | Batch |
|---|---|---|---|
| **TRA (CL)** | $\tilde{M}^\beta(s, s_+)/p^\beta(s_+)$ | $\sum_j p(\beta_j\vert s)\tilde{M}^{\beta_j}(s, s_+)/p^\beta(s_+)$ | $(s_t, s_+)^B, (s_i, s_j)^{B^2}$ |
| **TD-SR** | $\tilde{M}^\beta(s, s_+)/p^\beta(s_+)$ | $\tilde{M}^\beta(s, s_+)/p^\beta(s_+)$ | $(s_t, s_{t+1})^B, (s_i, s_j)^{B^2}$ |
| **BYOL** | $p^\beta(s_{t+1}\vert s_t)$ | $p^\beta(s_{t+1}\vert s_t)$ | $(s_t, s_{t+1})^B$ |
| **BYOL-$\gamma$** (ours) | $\tilde{M}^\beta(s, s_+)$ | $\sum_j p(\beta_j\vert s)\tilde{M}^{\beta_j}(s, s_+)$ | $(s_t, s_+)^B$ |

Table 1: **Auxiliary Representation Objectives**. We provide an overview of the representation objectives we consider. In the first two columns, we label the quantities which representations are approximating in finite MDPs, either from datasets with trajectories collected from a single policy $\tau \sim \beta$, or a mixture of policies $\tau \sim \{\beta_j\}$. We provide additional derivations for mixture datasets in Appendix F. In the last column, we list samples used for each objective, where the superscript denotes the number of loss terms for a pair of samples.

BYOL objective. Figure 1b depicts our overall representation learning objective. We can view this objective as iteratively minimizing an upper-bound on the error between $\psi(\phi((s))$ and a target of the true successor features of the policy $\psi^\pi$ with changing basis features $\bar{\phi}$. With convex $f$, by Jensen's inequality we have:

$$\mathcal{L}_{\text{BYOL-}\gamma} \geq \mathbb{E}_{s_t}\left[f(\psi(\phi(s_t)), \mathbb{E}_{s_+\sim\tilde{M}^\pi(s_t,s_+)}\bar{\phi}(s_+))\right] = \mathbb{E}_{s_t}\left[f(\psi(\phi(s_t)), (1-\gamma)\psi_{\bar{\phi}}^\pi(s_t))\right] \quad (9)$$

Specifically, we precisely show the relationship of BYOL-$\gamma$ to the SR with the following result:

**Theorem 4.1.** *Given a finite MDP with linear representations $\Phi \in \mathbb{R}^{\vert\mathcal{S}\vert\times d}$, and predictor $\Psi \in \mathbb{R}^{d\times d}$, under assumptions of orthogonal initialization for $\Phi$ (Ass. E.1), a uniform initial state distribution $p_0(s)$ (Ass. E.2), and symmetric transition dynamics (Ass. E.3), minimizing the self-predictive learning objective $\mathcal{L}_{BYOL\text{-}\gamma}(\phi, \psi)$ approximates a spectral decomposition of the successor representation $\tilde{M}^\pi \approx \Phi\Psi\Phi^T$, corresponding to successor features $(1-\gamma)\Psi^\pi \approx \Psi\Phi$.*

Proof is in Appendix E.2, where we show that existing theory (Khetarpal et al., 2025) also translates to the proposed BYOL-$\gamma$ objective. Finally, we can see the relation between this objective and CL (1), with the most striking difference being the removal of the denominator involving negative samples. Surprisingly, we reveal that this simplified system still captures similar information and can also lead to empirical generalization ( Section 5.1), while relying neither on TD learning nor negative samples.

**BYOL-$\gamma$ Variants.** We discuss a few variants on our base objective, namely, we evaluate **bidirectional prediction** (Guo et al., 2020; Tang et al., 2023) where we add an additional backwards predictor $\psi_b$ which predicts a past representation from the future. We also utilize an **action-conditioned** variant of the forward predictor $\psi_f(\phi(s_t), a_t)$, which can be interpreted as a temporally extended latent dynamics model, or capturing information about $\tilde{M}^\pi(s, a, s_+)$, giving:

$$\mathcal{L}_{\text{BYOL-}\gamma}(\phi, \psi) = \mathbb{E}_{s_t\sim\ p(s), s_+\sim\tilde{M}^\pi(s_t,s_+)}\left[f(\psi_f(\phi(s_t), a_t), \bar{\phi}(s_+)) + f(\bar{\phi}(s_t), \psi_b(\phi(s_+)))\right] \quad (10)$$

For $f$, we choose a cross-entropy loss between (softmax) normalized representations, similar to DINO (Caron et al., 2021): $f_{\text{CE}}(a, b) = \text{softmax}(b) \cdot \log\text{softmax}(a)$. However, we find that a normalized $l_2$ loss, $f_{l_2} = \|\frac{a}{\|a\|} - \frac{b}{\|b\|}\|_2^2$, commonly used in BYOL setups (Grill et al., 2020; Schwarzer et al., 2020) also works, which we ablate in Section 5.4.

## 4.2 TRAINING A POLICY WITH AUXILIARY REPRESENTATION

We consider BYOL-$\gamma$ and other objectives as auxiliary losses for BC policies $\pi_\Theta(a\vert s, g)$ to improve their generalization. We label all parameters $\Theta = (\theta, \phi, \psi)$, with the parameters of the encoder and predictor corresponding to $\phi, \psi$, respectively. The policy-head ($\theta$) transforms representations to actions via an MLP: $\pi_\Theta(a\vert s, g) = \text{MLP}_\theta(\text{concat}(\phi(s), \phi(g)))$. With this policy, we train with the objective:

$$\mathbb{E}_{\beta_j\sim p(\beta_j),\tau\sim\beta_j}\left[\mathcal{L}_{\text{BC}}(\Theta) + \alpha\mathcal{L}_{\text{aux}}(\phi, \psi)\right] \quad (11)$$

The term $\mathcal{L}_{\text{BC}}$ updates the parameters of both the policy head $\theta$ and its inputs, i.e., the encoder $\phi$, while $\mathcal{L}_{\text{aux}}$ updates $\psi, \phi$ but not $\theta$. With $\phi$ affected by both terms, the BC loss ensures that the

representation is sufficient for action prediction, preventing collapse, which can be an issue under certain representation learning objectives such as BYOL. Additionally, the auxiliary loss prevents overfitting and help generalization for the policy. We provide additional details about the architecture in Appendix A.

We wish to learn representations related to the successor measure of the mixture policy as motivated by Equation (7). However, there are trade-offs with the representation learning objectives in terms of the quantities they approximate and the data they use, as shown in Table 1. When we directly have full MC samples from a single policy ($\tau \sim \beta$), TRA, TD-SR, and BYOL-$\gamma$ each capture information related to its SM. However, in practice, we only have MC samples from individual policies $\tau \sim \{\beta_j\}$, rather than the mixture.

First, we consider using TD-SR as an auxiliary loss for BC, as we can see it still approximates the correct quantity. To our knowledge, we are the first to study this objective as an auxiliary loss for BC. TD-SR explicitly can "stitch" across policies, i.e. $\psi(s_t)\psi(s')$ via $\gamma\bar{\psi}(s_{t+1})\bar{\phi}(s')$ for $s_t, s_{t+1} \sim p^{\beta_i}(s_t)p^{\beta}(s_{t+1}|s_t)$ and $s' \sim p^{\beta_j}(s)$. However, we wish to understand if we can obtain *representations that help with generalization without TD*, as this may be more scalable when applied with policy learning. This leads us to quantify how MC methods behave when trained on mixture datasets.

While 1-step MC methods like BYOL are consistent across dataset composition, we can see that TRA and BYOL-$\gamma$ approximate different quantities than TD-SR. Namely, *rather than approximating the SM of the mixture policy, MC methods capture a mixture of SRs* as shown in Table 1 and Appendix F. In practice, MC methods still learn relationships between states encountered in different policies by effectively approximating many SRs in a single representation space, which is qualitatively shown in Figure 2. Surprisingly, we show that BYOL-$\gamma$ can lead to representations that are similar, or even better than TD-SR without utilizing TD. However, we find that CL as used in TRA leads to pessimism in the relationship between states sampled by different policies. Namely, for states that are not in the same trajectory, they will only be paired as negative examples, whose representations are pushed apart. This also shows up in the denominator of its approximation, with normalization from $p^{\beta}(s_+)$ in $\sum_j p(\beta_j|s)\tilde{M}^{\beta_j}(s, s_+)/p^{\beta}(s_+)$. On the other hand, this pessimism is not encountered with BYOL-$\gamma$, which does not utilize negative examples, giving an approximation of $\sum_j p(\beta_j|s)\tilde{M}^{\beta_j}(s, s_+)$ by simply predicting latents. Finally, we highlight that BYOL-$\gamma$ only computes $O(B)$ loss terms, while CL compute $O(B^2)$ negatives $(s_i, s_j)$, and we utilize $O(B^2)$ bootstrap terms with TD-SR.

## 5 EXPERIMENTS

Now that we have shown a theoretical basis for studying choices of representations, including **CL**, **TD-SR**, **BYOL**, and our new objective (**BYOL-$\gamma$**), we study how these methods behave empirically. We compare representation learning algorithms across three axes: (1) First, we compare qualitatively whether the representations appear to capture temporal relationships (2) Second, we assess representations quantitatively by measuring zero-shot generalization performance on unseen tasks that require combinatorial generalization (3) Third, we assess generalization performance over an increasing generalization horizon. Finally, we perform ablations on the various components of our proposed method to demonstrate the relative importance of each algorithmic choice.

**Environments.** We empirically evaluate how well our approach can help with combinatorial generalization on offline goal-reaching tasks on OGBench (Park et al., 2025), which contains both navigation and manipulation tasks, across low-dimensional and visual observations. We focus on navigation environments, where OGBench provides `stitch` datasets, that assess combinatorial generalization by training on trajectories that span at most 4 maze cells, while evaluating on tasks that are longer, requiring combining information from multiple smaller trajectories.

**Baselines.** We benchmark against non-hierarchical methods that perform control from state to low-level actions (e.g. joint-control). In addition to **BYOL-$\gamma$** used as an auxiliary loss for BC, we evaluate several baselines: **GCBC** is the standard BC baseline, which we aim to improve upon with representation learning. **Offline RL** from OGBench, including implicit {V,Q}-learning (**IVL, IQL**) (Kostrikov et al., 2022), Quasimetric RL (**QRL**)(Wang et al., 2023), and Contrastive RL (**CRL**) (Eysenbach et al., 2022). **BYOL** is a minimal version of our setup with 1-step prediction ($\gamma = 0$), only forwards prediction ($\psi_f$) without action-conditioning ($\psi_f(\phi(s_t))$), and loss $f_{l_2}$. **TRA**

Figure 2: **Visualization of the Learned Representation: depicts the similarity between the prediction of the current state representation to the goal representation.** For **BYOL-γ** and **TD-SR**, we visualize the cosine similarity between $\psi(\phi(s), \cdot)$ or $\psi(s, \cdot)$, to $\phi(g)$ $\forall s \in D$ for a fixed goal $g$ which is indicated by the star marked in red.

(Myers et al., 2025b) is an auxiliary representation objective using contrastive learning related to an MC approximation of the SM. **TD-SR** is a TD-based approximation of the SM as in Equation (2) used as an auxiliary objective for BC. We also compare to an $n$-step version of BYOL in Appendix B.2 and the Forward-Backward (**FB**) (Touati and Ollivier, 2021; Touati et al., 2023) in Appendix B.3.

**Experimental Setup.** We match the training details of OGBench, and consider a similar representation learning setup to **TRA**. We found it was beneficial to add action conditioning to **TD-SR**, but did not see an overall improvement for **TRA**, so we use the original setup without action-conditioning. While we use policy $\pi(\phi(s), \phi(g))$ and train with action-conditioning for **BYOL-γ** and **TD-SR**, **TRA** originally uses a parameterization $\pi(\psi(s), \phi(g))$ and does not condition on actions. We provide a full comparison for changing $\psi(s)$ to $\phi(s)$ and action-conditioning in Appendix B.1 for **TRA**. Howeve,r we obtain similar performance on average with the original setup. For clarity, in Table 2, we utilize superscript $a$ to denote methods with action-conditioning. Notably, we find that the weight of the auxiliary representation learning objectives ($\alpha$) can be sensitive to both the embodiment, and size of environment (`medium` vs `large`). For each method, we perform a hyperparameter sweep over 4 $\alpha$ values, and report the best result for each environment in Table 2. We hold other hyperparameters constant, except with variation between non-visual and visual noted in Appendix A.

## 5.1 QUALITATIVE ANALYSIS OF REPRESENTATIONS

In Figure 2, we display a qualitative analysis of the representations. We visualize the similarity between the future prediction $\psi$ for each state to $\phi(g)$ for a fixed goal $g$. We can see that **BYOL-γ** seems to learn a representation that encodes reachability between states, and has a similar structure to **TD-SR**, which is known to approximate the successor measure. **TRA** and base **BYOL** seem to both capture similar structure and learn a less well-defined latent space. However, **BYOL-γ** and **TD-SR** have more distinct similarity, and have visible "paths" of similar states. **BYOL-γ** also appears to capture the most similarity among more distant pairs of states. Compared to **TRA**, our hypothesis here is that **BYOL-γ** has more optimistic similarity between distant states due to the lack of a negative term in the loss, pushing representations apart. We show additional environments in Appendix H.1. We also check the correlation of distance in representation space with shortest paths in the maze in Appendix H.2, showing that **BYOL-γ** best captures the structure of the environment.

## 5.2 ZERO-SHOT PERFORMANCE ON COMBINATORIAL GENERALIZATION TASKS

In Table 2, we provide the performance results across all methods. Overall, our proposed method **BYOL-γ**, shows improved performance vs. **GCBC** across most environments, and is either competitive with or outperforms **TD-SR** and **TRA**. Importantly, we find that a minimal **BYOL** setup does not confer significant benefit over the base **GCBC** except in non-visual `antmaze` environments. Generally, auxiliary representation learning with **GCBC** outperforms existing offline RL methods.

Within the auxillary loss methods, we find that **TD-SR** and **BYOL-γ** tend to outperform **TRA** on most environments. While we find that **TD-SR** outperforms **BYOL-γ** on environments with smaller state spaces (`antmaze-{medium, large}`), we find that **BYOL-γ**'s simpler training procedure is beneficial in environments with larger state spaces (`humanoidmaze-{medium, large}`, `visual-antmaze-medium` and `visual-scene-play`).

| Dataset | BYOL-$\gamma^a$ | BYOL | TRA | TD-SR$^a$ | GCBC | GCIVL | GCIQL | QRL | CRL |
|---|---|---|---|---|---|---|---|---|---|
| antmaze-medium-stitch | $58 \pm 5$ | $59 \pm 4$ | $54 \pm 6$ | $64 \pm 6$ | $45 \pm 11$ | $44 \pm 6$ | $29 \pm 6$ | $59 \pm 7$ | $53 \pm 6$ |
| antmaze-large-stitch | $19 \pm 7$ | $17 \pm 6$ | $11 \pm 8$ | $23 \pm 4$ | $3 \pm 3$ | $18 \pm 2$ | $7 \pm 2$ | $18 \pm 2$ | $11 \pm 2$ |
| humanoidmaze-medium-stitch | $51 \pm 6$ | $23 \pm 3$ | $45 \pm 8$ | $42 \pm 4$ | $29 \pm 5$ | $12 \pm 2$ | $12 \pm 3$ | $18 \pm 2$ | $36 \pm 2$ |
| humanoidmaze-large-stitch | $13 \pm 3$ | $3 \pm 1$ | $5 \pm 4$ | $11 \pm 3$ | $6 \pm 3$ | $1 \pm 1$ | $0 \pm 0$ | $3 \pm 1$ | $4 \pm 1$ |
| antsoccer-arena-stitch | $25 \pm 5$ | $12 \pm 7$ | $14 \pm 4$ | $22 \pm 10$ | $24 \pm 8$ | $21 \pm 3$ | $2 \pm 0$ | $1 \pm 1$ | $1 \pm 0$ |
| visual-antmaze-medium-stitch | $68 \pm 4$ | $57 \pm 8$ | $52 \pm 3$ | $49 \pm 2$ | $67 \pm 4$ | $6 \pm 2$ | $2 \pm 0$ | $0 \pm 0$ | $69 \pm 2$ |
| visual-antmaze-large-stitch | $26 \pm 5$ | $26 \pm 5$ | $17 \pm 1$ | $29 \pm 2$ | $24 \pm 3$ | $1 \pm 1$ | $0 \pm 0$ | $1 \pm 1$ | $11 \pm 3$ |
| visual-scene-play | $17 \pm 1$ | $13 \pm 3$ | $16 \pm 3$ | $14 \pm 1$ | $12 \pm 2$ | $25 \pm 3$ | $12 \pm 2$ | $10 \pm 1$ | $11 \pm 2$ |
| average-state | **33** | 23 | 26 | **32** | 21 | 19 | 10 | 20 | 21 |
| average-visual | **37** | 32 | 28 | 31 | 34 | 11 | 5 | 4 | 30 |
| average-all | **35** | 26 | 27 | 32 | 26 | 16 | 8 | 14 | 25 |

Table 2: **OGBench: We find that BYOL-$\gamma$ performs better overall compared to prior methods**. We report mean and standard deviation over 10 training seeds in non-visual environments, and 4 seeds in visual environments. We match the OGBench evaluation setup of 5 evaluation (state,goal) tasks, and 50 episodes per task. The success rate is then averaged over the last 3 checkpoints. We color the best non-RL method, and **bold** values within 95% of its value in the same row. We use superscript $a$ to denote methods utilizing action-conditioning.

Interestingly, in `visual-antmaze` **TRA** and **TD-SR** actually seem to hurt performance in comparison to base **GCBC**. On the other hand, with **BYOL-$\gamma$** we see no performance degradation over **GCBC** on the visual environments, a considerable improvement over other methods. In Appendix C, we extend **BYOL-$\gamma$** to the hierarchical setting (**HBYOL-$\gamma$**), where we obtain significant improvement over BC baselines, including on visual maze environments.

We also find a relationship between success and representation quality (Section 5.1). Namely, in Table 12 we calculate the correlation of representations to shortest path distances and success rate over these same checkpoints. We see that the ranking of methods in terms of average correlation to shortest path (`average maze correlation`) in representation space matches the ordering of methods in terms of average empirical policy success (`average maze success`).

## 5.3 EVALUATING GENERALIZATION WITH INCREASING HORIZON

We conduct experiments to understand how success rate changes as an agent has to reach more challenging goals further away from its starting position. For each `maze` environment, we consider the same base 5 evaluation tasks used in Table 2, but construct intermediate waypoints along the shortest path to the final goal determined by breadth-first search. We also include an additional `maze` environment, `giant` on which all methods have zero success rates to reach distant goals. This gives a more holistic view on an agent's performance.

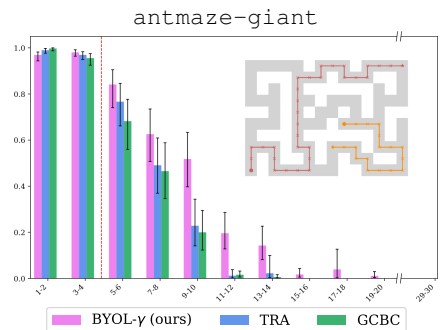

Figure 3: **Evaluating Generalization with Increasing Horizons**: shows that **BYOL-$\gamma$** not only performs well on goals in the near horizon, but also, helps to generalize well to goals requiring stitching, after the red bar ($> 4$).

We display results in Figure 3 and Appendix G, where we can see how performance drops off for all methods after a generalization threshold denoted by the red bar. While all methods cannot fully reach distant goals on `giant`, we see that **BYOL-$\gamma$** has the slowest drop-off in performance. We note that this is a challenging task, that requires stitching up to approximately 8 different trajectories.

## 5.4 COMPONENTS AFFECTING GENERALIZATION

We ablate key components of the **BYOL-$\gamma$** objective in Table 3. This includes removing action conditioning for forward predictor $\psi_f$ $(-a)$, swapping the loss from cross-entropy to normalized squared $l_2$ norm $(f_{l_2})$, removing backwards predictor $\psi_b$, and predicting the representation of the adjacent state $(\gamma = 0)$. Both removing action-conditioning, and backwards prediction overall lead to

Table 3: **BYOL-$\gamma$ ablations.** For each ablation, we perform sweep over $\alpha$, and report the best result per-environment. For all environments, we report results over 4 seeds (for **BYOL-$\gamma$**, we use the first 4 of 10 in Table 2).

| Dataset | BYOL-$\gamma^a$ | $-a$ | $f_{l_2}$ | $-\psi_b$ | $\gamma = 0$ |
|---|---|---|---|---|---|
| antmaze-medium-stitch | $61 \pm 6$ | $63 \pm 9$ | $56 \pm 4$ | $67 \pm 2$ | $59 \pm 5$ |
| antmaze-large-stitch | $21 \pm 5$ | $27 \pm 7$ | $24 \pm 6$ | $19 \pm 7$ | $8 \pm 4$ |
| humanoidmaze-medium-stitch | $54 \pm 5$ | $48 \pm 5$ | $49 \pm 6$ | $52 \pm 5$ | $18 \pm 2$ |
| humanoidmaze-large-stitch | $14 \pm 2$ | $12 \pm 6$ | $15 \pm 7$ | $13 \pm 2$ | $3 \pm 1$ |
| antsoccer-arena-stitch | $21 \pm 4$ | $20 \pm 5$ | $11 \pm 5$ | $27 \pm 7$ | $25 \pm 7$ |
| visual-antmaze-medium | $68 \pm 4$ | $65 \pm 3$ | $63 \pm 5$ | $61 \pm 4$ | $54 \pm 9$ |
| visual-antmaze-large | $26 \pm 5$ | $25 \pm 8$ | $27 \pm 7$ | $28 \pm 2$ | $28 \pm 1$ |
| average-all | $33$ | $33$ | $31$ | $33$ | $24$ |

similar results, but variability per-environment. For $f_{l2}$, we obtain slightly worse average performance, and for $\gamma = 0$, we see the largest drop-off, especially on `humanoidmaze`.

## 6 DISCUSSION

**Limitations.** While we demonstrate that BYOL-$\gamma$ and other representation learning objectives offer a promising recipe for obtaining combinatorial generalization, we find that there still exists a generalization gap, especially on challenging navigation environments e.g. `giant`. We also find a less significant improvement over BC on visual environments, which may motivate additional investigation. Additionally, we may anticipate more benefit from representation learning when applied to larger visual datasets, which has been fruitful in other domains.

**Conclusion.** In this work, we provide a stronger understanding of the relationship between quantities related to successor representations and the generalization of policies trained with behavioral cloning through a unified understanding of objectives. We propose a new self-predictive representation learning objective, BYOL-$\gamma$, and show that it captures information related to the successor measure, resulting in a competitive choice of an auxiliary loss for better generalization. We demonstrate that augmenting behavior cloning with meaningful representations results in new capabilities such as improved combinatorial generalization, especially in larger and more complex environments.

## 7 ACKNOWLEDGMENTS

The authors thank Zhaohan Daniel Guo, Faisal Mohamed, and Özgür Aslan for feedback on earlier drafts of the work.

We want to acknowledge funding support from Natural Sciences and Engineering Research Council of Canada, Samsung AI Lab, Google Research, Fonds de recherche du Québec and The Canadian Institute for Advanced Research (CIFAR) and compute support from Digital Research Alliance of Canada, Mila IDT, and NVidia.

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

## A EXPERIMENTAL SETUP

Table 4: Hyperparameters for BYOL-$\gamma$

| Hyperparameter | Shared | |
|---|---|---|
| actor head | MLP (512,512,512) | |
| representation encoder ($\phi$) | MLP (64,64,64) | |
| predictor ($\psi$) | MLP (64,64,64) | |
| encoder ensemble | 2 | |
| learning rate | $3 \times 10^{-4}$ | |
| optimizer | Adam | |

| | Non-visual | Visual |
|---|---|---|
| Gradient steps | 1000k | 500k |
| Batch size | 1024 | 256 |
| $\tau$ (EMA) | 1.0 | 0.99 |
| $\gamma$ | 0.99 | {0.66, 0.99} |
| $\alpha$ (alignment) | {1,6,40,100} | {1,6,10,20} |
| additional encoder | n/a | impala_small |
| encoder output dimension | $|s|$ | 64 |

### A.1 IMPLEMENTATION DETAILS

In this section we provide more training details for BYOL-$\gamma$, and representation learning baselines. We match the training details of OGBench, including gradient steps, batch size, learning rate.

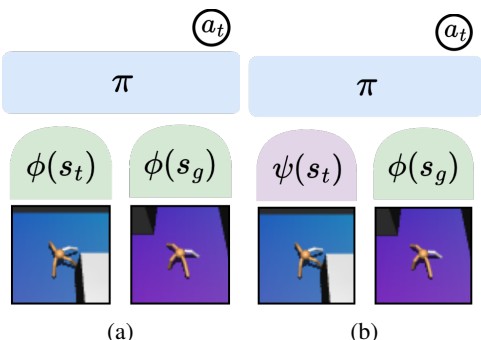

(a)          (b)

**Network Architecture.** We follow the same general setup as TRA, where we utilize MLP-based encoders, and action head. For the output dimension of the encoder, we use the state dimension for non-visual experiments, and $64$ for visual experiments. For the predictor $\psi$, we utilize an MLP of the same architecture as the encoder. For image-based tasks, there is an additional CNN, which then passes output to the MLP encoder.

Figure 4: **Encoder Variation**. When training with BYOL, BYOL-$\gamma$ and TD-SR, we utilize policies with architecture (a) which uses $\phi$ to process states and goals. We utilize architecture (b) for TRA to match prior implementation, however in Appendix B.1 we train TRA with architecture (a) and action-conditioning.

**Representation Ensemble.** We follow the setup of TRA which utilizes representation ensembling, such that two copies of the encoder $\phi_1$, $\phi_2$ are in parallel. We also have two distinct predictors $\psi_1$, $\psi_2$ for each ensemble. As input to the policy head, we average the representations, $\bar{z} = \frac{\phi_1(s_t) + \phi_2(s_2)}{2}$. Each representation is trained independently for the BYOL loss, but the BC loss differentiates through both $\phi$s.

**Alignment.** We find that the choice of weight of the auxiliary loss for the representation learning objective is sensitive to both the robot embodiment and the environment size. For comparison, we perform a hyperparameter search over four alignment values for BYOL-$\gamma$, TRA, and TD-SR, and then report the best value for each environment in Table 2.

**Discount.** For sampling the next-state, we utilize a discount factor of $\gamma = 0.99$ for all non-visual environments. For visual environments, we perform a hyperparameter search over $\{0.66, 0.99\}$, however all representation learning methods performed better at $\gamma = 0.66$.

## A.2 BYOL-$\gamma$

**Target network.** For BYOL, we find that exponential moving average (EMA) target networks for the encoder $\phi$ are not necessary for non-visual environments ($\tau = 1$), but for visual environments, we find that a fast target stabilizes training ($\tau = 0.99$):

$$\phi_{\text{target}} = \tau\phi_{\text{online}} + (1 - \tau)\phi_{\text{target}}$$

## A.3 TRA

In practice, TRA uses a symmetric version (Radford et al., 2021) of the InfoNCE objective discussed in Equation 1. We write this in batch form, $\mathcal{B} = \{(s_i, s_{+,i})\}_{i=1}^{|\mathcal{B}|}$ rather than in expectation:

$$\mathcal{L}_{\text{TRA}} = \mathbb{E}_{\mathcal{B}}\left[ -\frac{1}{B}\sum_{i=1}^{|\mathcal{B}|} \log \frac{e^{f(\psi(s_i),\phi(s_{+,i}))}}{\sum_{j=1}^{|\mathcal{B}|} e^{f(\psi(s_i),\phi(s_{+,j}))}} - \frac{1}{B}\sum_{i=1}^{|\mathcal{B}|} \log \frac{e^{f(\psi(s_i),\phi(s_{+,i}))}}{\sum_{j=1}^{|\mathcal{B}|} e^{f(\psi(s_j),\phi(s_{+,i}))}} \right] \quad (12)$$

Additionally, TRA minimizes the squared norm of representations $\min_{\phi,\psi} \lambda\mathbb{E}_s\left[\frac{\|\phi(s)\|^2}{d} + \frac{\|\psi(s)\|^2}{d}\right]$ with $\lambda = 10^{-6}$. For TRA, we search over $\alpha = \{10, 40, 60, 100\}$.

## A.4 TD-SR

Prior work similar to TD-SR, which trains FB for zero-shot policy optimization (Touati et al., 2023) typically normalizes $\phi$ with an additional loss term so that $\mathbb{E}\left[\phi\phi^T\right] \approx I_d$. However, we found that adding this loss term was not beneficial to performance in our setting and hence do not include it.

TD-SR uses an EMA target network as described in A.2 with $\tau = 0.005$. For TD-SR, we search over $\alpha = \{0.01, 0.05, 0.001, 0.005\}$.

## A.5 CODE.

We utilize the OGBench (Park et al., 2025) codebase and benchmark, and its extensions in the TRA codebase (Myers et al., 2025b) for equal comparison.

## A.6 COMPUTE REQUIREMENTS

We perform all experiments utilizing single GPUs, predominately NVIDIA RTXA8000 and L40S. We utilize 6 CPU cores, 24G of RAM for non-visual environments, and 64G for visual experiments. Experiments take 2 to 4 hours for non-visual and 6 to 12 hours for visual environments.

## B ABLATIONS.

### B.1 ACTION-CONDITIONING

In this section, we ablate the component of performing action-conditioning for the predictor $\psi(s_t)$ vs $\psi(s_t, a_t)$ for TRA and TD-SR. We consider a similar comparison for BYOL-$\gamma$ in Table 3. For this comparison, when we perform action-conditioning, we utilize a policy representation $\pi(s = \phi(s), g = \phi(g))$ as we have predictor $\psi(s, a)$, and otherwise $\pi(s = \psi(s), g = \phi(g))$ as in the original TRA implementation. We find that results can be environment specific. On average, results are not improved for TRA, but we find an improvement for TD-SR, hence in our main Table 2 we include the action-conditioned results for TD-SR and the action-free results for TRA to match the original implementation.

| Dataset | TRA | TRA$^a$ | TD-SR | TD-SR$^a$ |
|---|---|---|---|---|
| `antmaze-medium-stitch` | $54 \pm 6$ | $57 \pm 12$ | $64 \pm 10$ | $64 \pm 6$ |
| `antmaze-large-stitch` | $11 \pm 8$ | $7 \pm 7$ | $17 \pm 6$ | $23 \pm 4$ |
| `humanoidmaze-medium-stitch` | $45 \pm 8$ | $45 \pm 5$ | $36 \pm 3$ | $42 \pm 4$ |
| `humanoidmaze-large-stitch` | $5 \pm 4$ | $9 \pm 4$ | $6 \pm 2$ | $11 \pm 3$ |
| `antsoccer-arena-stitch` | $14 \pm 4$ | $25 \pm 8$ | $17 \pm 5$ | $22 \pm 10$ |
| `visual-antmaze-medium-stitch` | $52 \pm 3$ | $33 \pm 4$ | $47 \pm 5$ | $49 \pm 2$ |
| `visual-antmaze-large-stitch` | $17 \pm 1$ | $22 \pm 5$ | $28 \pm 3$ | $29 \pm 2$ |
| `visual-scene-play` | $16 \pm 3$ | $18 \pm 2$ | $12 \pm 2$ | $14 \pm 1$ |
| `average-all` | 27 | 27 | 28 | 32 |

Table 5: **Action-conditioning ablations.** We ablate the choice to condition on the first action for predictor $\psi$ for TRA and FB over 10 seeds for non-visual and 4 seeds for visual environments.

| Dataset | BYOL-$\gamma^a$ | $-\psi_b$ | $\gamma = 0$ | $-\psi_b, \gamma = 0$ | BYOL$^a_{n=1}$ | BYOL$^a_{n=3}$ | BYOL$^a_{n=5}$ |
|---|---|---|---|---|---|---|---|
| `antmaze-medium-stitch` | $61 \pm 6$ | $67 \pm 2$ | $59 \pm 5$ | $60 \pm 5$ | $60 \pm 8$ | $60 \pm 7$ | $58 \pm 7$ |
| `antmaze-large-stitch` | $21 \pm 5$ | $19 \pm 7$ | $8 \pm 4$ | $13 \pm 5$ | $19 \pm 4$ | $8 \pm 4$ | $3 \pm 3$ |
| `humanoidmaze-medium-stitch` | $54 \pm 5$ | $52 \pm 5$ | $18 \pm 2$ | $27 \pm 7$ | $33 \pm 4$ | $32 \pm 2$ | $20 \pm 1$ |
| `humanoidmaze-large-stitch` | $14 \pm 2$ | $13 \pm 2$ | $3 \pm 1$ | $5 \pm 2$ | $3 \pm 2$ | $3 \pm 1$ | $5 \pm 2$ |
| `antsoccer-arena-stitch` | $21 \pm 4$ | $27 \pm 7$ | $25 \pm 7$ | $23 \pm 6$ | $25 \pm 12$ | $11 \pm 7$ | $13 \pm 9$ |
| `average-all` | 34 | 36 | 23 | 26 | 28 | 23 | 20 |

Table 6: **N-step BYOL ablations**. We ablate BYOL-$\gamma$ with an n-step BYOL baseline, where we report results over 4 seeds.

## B.2 N-STEP BYOL

We provide an additional BYOL-based baseline that utilizes $n$-step next-representation recurrent prediction, while BYOL-$\gamma$ uses non-recurrent prediction. We utilize the same BYOL-$\gamma$ architecture with forward prediction, but with the following objective, computing loss with $n$ terms, where $\psi_f^n$ is shorthand for $n$ recurrent calls:

$$\mathcal{L} = f(\psi_f(\phi(s_t, a_t), \bar{\phi}(s_{t+1})) + f(\psi_f(\psi_f(\phi(s_t, a_t), a_{t+1}), \bar{\phi}(s_{t+2})) + \cdots + f(\psi_f^n(\cdot), \bar{\phi}(s_{t+n}))$$

**Theoretically,** in a finite MDP, we can interpret this objective of capturing information up to $n$-step transitions (Tang et al., 2023), i.e. information related to $\{P_a, P_a^2, \cdots, P_a^n\}$ is captured by loss terms $\{\psi_f, \psi_f^2, \cdots, \psi_f^n\}$ respectively. As BYOL-$\gamma$ captures information related to $\tilde{M}^\pi = (1 - \gamma) \sum_{t \geq 0} \gamma^t P_\pi^t$, these two objectives match in theory at $\gamma = 0$. In practice, as $n$-step operates recurrently, we are constrained to a shorter $n$ which limits the ability for learning long-horizon information.

**Empirically,** we report comparison of $n$-step with $n = \{1, 3, 5\}$ to BYOL-$\gamma$ in Table 6. We validate our n-step implementation in the base case ($n = 1$) with the ablation $\{-\psi_b, \gamma = 0\}$ to BYOL-$\gamma$ making them equivalent. With increased multi-step prediction (as we increase $n$), we find worse performance on average.

## B.3 FORWARD-BACKWARD ALGORITHM

As an alternative to GCBC, we could instead consider the full Forward-Backward (FB) algorithm for zero-shot goal-reaching, as proposed in Touati et al. (2023). Here, instead of conditioning the policy on a goal representation $\phi(g)$, we instead condition $\psi$ on a vector $z$ such that jointing learning $\phi$ and $\psi$ produces a *policy-dependent* successor representation where

$$M^{\pi_z}(s, a, s_+) = \psi(s, a, z)^\top \phi(s_+) \cdot p(s_+), \quad \text{and} \quad \pi_z(a \mid s) := \underset{a}{\operatorname{argmax}} F(s, a, z)^\top z, \quad (13)$$

$\psi$ and $\phi$ can be learned through a TD relationship analogous to Equation 2, additionally sampling vectors $z$ according to some distribution. In the discrete setting, the policy can be derived directly from Equation 13. In the continuous setting, Touati and Ollivier (2021) additionally learn a policy network $\pi(s, z)$, trained to maximize $F(s, a, z)^\top z$, in a DDPG-style (Lillicrap et al., 2015) algorithm.

At inference time, a policy for a goal state $g$ can be obtained by first encoding the goal state to the $z$-representation space using the relationship $z = \mathbb{E}_{s \sim \beta} [r(s)\phi(s)]$, which implies $z = \phi(g)$ for goal-reaching tasks.

For these experiments, we follow the $z$ sampling method from Touati and Ollivier (2021) by using a 50-50 mixture of states $s$ sampled from $\beta$ and encoded to $z = \phi(s)$ and vectors sampled uniformly on a sphere of radius $\sqrt{d}$ where $d$ is the latent dimension. We use network architectures for $\phi$ and $\psi$ matching those used in the implementation of FB provided in Tirinzoni et al. (2025), however we keep the number and size of the hidden layers as well as the latent dimension consistent with our implementations of other methods. Additionally, we add a BC-loss to the policy loss as a regularization, with coefficient 1. We sweep the learning rate over three values: $\{10^{-4}, 10^{-5}, 10^{-6}\}$ and selected the best performing, averaged over four seeds, for each environment.

In Table 7 we compare our proposed BC with auxiliary loss methods (**TD-SR**$^a$ and **BYOL-**$\gamma^a$), which use successor measure learning as an auxiliary loss for BC, to value-based methods which instead use a goal-conditioned value function (**GCIQL**) or successor measure (**FB**) to learn a policy through RL. We find that auxiliary loss methods significantly outperform across almost all environments.

| Dataset | BYOL-$\gamma^a$ | TD-SR$^a$ | FB | GCIQL |
|---|---|---|---|---|
| antmaze-medium-stitch | $61 \pm 6$ | $\mathbf{64} \pm 6$ | $36 \pm 5$ | $29 \pm 6$ |
| antmaze-large-stitch | $21 \pm 5$ | $22 \pm 3$ | $5 \pm 4$ | $7 \pm 2$ |
| humanoidmaze-medium-stitch | $\mathbf{54} \pm 5$ | $41 \pm 5$ | $26 \pm 5$ | $12 \pm 3$ |
| humanoidmaze-large-stitch | $\mathbf{14} \pm 2$ | $12 \pm 3$ | $2 \pm 1$ | $0 \pm 0$ |
| antsoccer-arena-stitch | $\mathbf{21} \pm 4$ | $18 \pm 12$ | $19 \pm 4$ | $2 \pm 0$ |
| average-all | $\mathbf{34}$ | $31$ | $17$ | $10$ |

Table 7: **Forward-Backward Algorithm**. We compare our proposed GCBC with auxiliary loss methods (**TD-SR**$^a$ and **BYOL-**$\gamma^a$) to an implementation of the Forward-Backward algorithm (**FB**) and an offline RL method **GCIQL**, both of which learn an actor which maximizes a goal-conditioned value function. We report best results from a hyperparameter sweep, averaged over four seeds

### B.4 Constant Encoder Output Dimension

Our main experimental setup utilized in Table 2 and other experiments utilize an encoder output dimension of size equal to the state dimension, corresponding to ant $|s| = 29$, and humanoid $|s| = 69$ for non-visual environments. We perform an additional comparison in Table 8 using a fixed size latent dimension $= 64$, matching the latent dimension used for visual environments. We can see that the larger latent dimension helps performance for each method on antmaze. Generally, we see similar trends to our prior experiments, such as BYOL-$\gamma$ performing stronger in humanoidmaze experiments, while FB performs stronger on antmaze.

| Dataset | BYOL-$\gamma^a$ | TD-SR$^a$ | TRA | TRA (Myers et al., 2025b) |
|---|---|---|---|---|
| antmaze-medium-stitch | $64 \pm 7$ | $\mathbf{73} \pm 8$ | $67 \pm 6$ | $61 \pm 3$ |
| antmaze-large-stitch | $18 \pm 7$ | $\mathbf{24} \pm 9$ | $15 \pm 10$ | $13 \pm 2$ |
| humanoidmaze-medium-stitch | $\mathbf{48} \pm 7$ | $41 \pm 3$ | $41 \pm 5$ | $46 \pm 2$ |
| humanoidmaze-large-stitch | $\mathbf{12} \pm 5$ | $10 \pm 2$ | $4 \pm 2$ | $9 \pm 1$ |
| antsoccer-arena-stitch | $\mathbf{21} \pm 10$ | $12 \pm 4$ | $18 \pm 5$ | $17 \pm 1$ |
| average-all | $\mathbf{33}$ | $\mathbf{32}$ | $29$ | $29$ |

Table 8: **Constant Encoder Output Dimension**. We conduct an ablation repeating our experimental setup for representation learning methods with a constant encoder output dimension at $64$. For reference, we also report results from Myers et al. (2025b).

### B.5 GCBC Encoder Ablation

We perform an ablation where we use the same architecture as representation learning methods for GCBC. Standard GCBC learns a shared state-goal encoder, $\phi(s, g)$, while representation learning methods pass inputs through an encoder separately, $\phi(s)$, $\phi(g)$ and representations are concatenated

and fed to an action head. In our main results, we report OGBench GCBC results with shared state-goal encoder, as this is a stronger baseline. However, to better illustrate the impact that auxiliary loss learning has on GCBC performance, in Table 9, we report GCBC results for an architecture matching representation learning methods (GCBC-$\phi$). We especially see a difference in visual environments, where state, goal are stacked ($64 \times 64 \times 6$) before going through the CNN.

| Dataset | GCBC | GCBC-$\phi$ |
|---|---|---|
| antmaze-medium-stitch | $45 \pm 11$ | $33 \pm 5$ |
| antmaze-large-stitch | $3 \pm 3$ | $5 \pm 4$ |
| humanoidmaze-medium-stitch | $29 \pm 5$ | $32 \pm 6$ |
| humanoidmaze-large-stitch | $6 \pm 3$ | $4 \pm 3$ |
| visual-antmaze-medium-stitch | $67 \pm 4$ | $37 \pm 6$ |
| visual-antmaze-large-stitch | $24 \pm 3$ | $4 \pm 3$ |
| visual-scene-play | $12 \pm 2$ | $10 \pm 1$ |
| average-state | $21$ | $\mathbf{19}$ |
| average-visual | $34$ | $17$ |
| average-all | $27$ | $18$ |

Table 9: **GCBC Encoder Ablation.**

## C  HIERARCHICAL POLICIES

Although we focus on the impact of representation learning in "flat" learning methods, hierarchical policies are also an effective orthogonal direction for improving generalization to longer horizon tasks. We demonstrate that BYOL-$\gamma$ also improves on a hierarchical GCBC setup (HGCBC) used in Frans et al. (2025). With HGCBC, we train a high-level policy $\pi^h(l \,|\, s, g)$ that predicts sub-goals $l$, and a low-level policy conditioned on sub-goals $\pi^h(a \mid s, l)$, which are both trained with BC. Using BYOL-$\gamma$, we implement its hierarchical version, **HBYOL-$\gamma$**, as follows: (1) perform our standard BYOL-$\gamma$ setup, which produces $\pi^l(a \mid \phi(s), \phi(l))$ and (2) train a hierarchical policy in the existing latent space of the low-level policy $\pi^h(\bar\phi(l) \mid \bar\phi(s), \bar\phi(g))$, where $\bar\phi$ denotes that the representation is fixed for the high-level policy. **HBYOL-$\gamma$** allows for both policies to operate in a shared representation space, and avoids state reconstruction performed by standard HGCBC. Prior work does not implement HGCBC in visual settings which would require predicting in pixel space. As a baseline, we implement HGCBC-$\phi$, which avoids pixel prediction by using GCBC-$\phi$ (Appendix B.5). This matches our HBYOL-$\gamma$ architecture, and two-stage setup but without representation learning on $\phi$. For HGCBC-$\phi$, we also found it was better to only use representations $\phi$ for the output space of $\pi_h$, and to train a shared input encoder from scratch.

| Dataset | GCBC | HGCBC | HGCBC-$\phi$ | BYOL-$\gamma^a$ | HBYOL-$\gamma^a$ | HIQL |
|---|---|---|---|---|---|---|
| antmaze-medium-stitch | $45 \pm 11$ | $60 \pm 4$ | · | $61 \pm 6$ | $76 \pm 12$ | $94 \pm 1$ |
| antmaze-large-stitch | $3 \pm 3$ | $11 \pm 8$ | · | $21 \pm 5$ | $29 \pm 9$ | $67 \pm 5$ |
| humanoidmaze-medium-stitch | $29 \pm 5$ | $35 \pm 4$ | · | $54 \pm 5$ | $61 \pm 2$ | $88 \pm 2$ |
| humanoidmaze-large-stitch | $6 \pm 3$ | $4 \pm 0$ | · | $14 \pm 2$ | $21 \pm 3$ | $28 \pm 3$ |
| visual-antmaze-medium-stitch | $67 \pm 4$ | · | $74 \pm 6$ | $68 \pm 4$ | $84 \pm 8$ | $87 \pm 2$ |
| visual-antmaze-large-stitch | $24 \pm 3$ | · | $19 \pm 1$ | $26 \pm 5$ | $31 \pm 3$ | $28 \pm 2$ |
| visual-scene | $12 \pm 2$ | · | $8 \pm 3$ | $17 \pm 1$ | $14 \pm 2$ | $49 \pm 4$ |
| average-state | $21$ | $28$ | · | $38$ | $47$ | $69$ |
| average-visual | $34$ | · | $34$ | $37$ | $43$ | $55$ |
| average-all | $27$ | · | · | $37$ | $45$ | $63$ |

Table 10: **Hierarchical BC with BYOL-$\gamma$.** We report performance averaged over 4 seeds, for HGCBC, HGCBC$-\phi$, and (H)BYOL-$\gamma$. We report GCBC and HIQL results from OGBench. We **highlight** the the best performing BC methods, **bold** for methods within $95\%$ of the BC, and **darker highlight** for BC methods which are within $95\%$ or better than HIQL.

In Table 10, we compare between these BC setups and also report results of hierarchical implicit Q-learning (HIQL) (Park et al., 2023). For HGCBC methods, we use a sub-goal ($l$) step of 25, listing other hyperparamters in Table 11. We see that HBYOL-$\gamma$ is the strongest BC setup, outperforming non-hierarchical BYOL-$\gamma$ and HGCBC. HBYOL-$\gamma$ is also competitive with HIQL, especially on visual-antmaze environments.

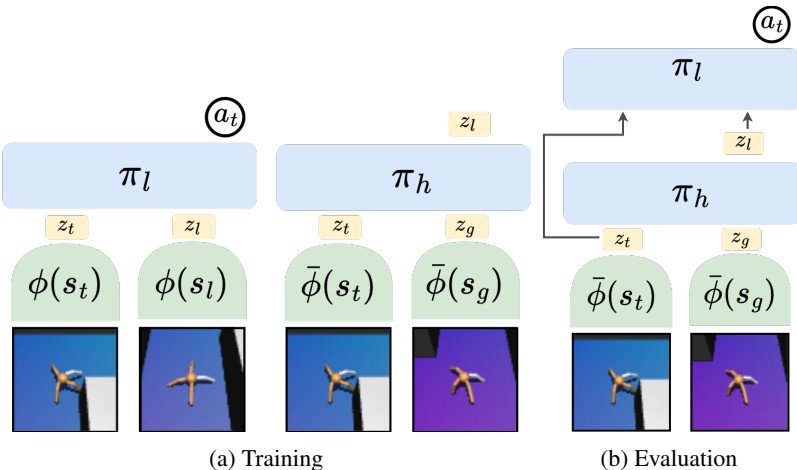

(a) Training        (b) Evaluation

Figure 5: **Architecture for HBYOL-$\gamma$.** During training (a), we first train a low-level policy with BYOL-$\gamma$. Then, we train a high-level policy with a similar procedure to $\pi_h$ in HGCBC, but using the representation space $\phi$. To train $\pi_h$, we freeze $\phi$, labeled $\bar{\phi}$, and use it for encoding inputs, and the output space, where the $\pi_h$ predicts the representation of the sub-goal: $z_l = \bar{\phi}(s_l)$. During evaluation (b), $\pi_h$ first predicts a sub-goal representation, which is then passed to the $\pi_h$, where both policies utilize a common state representation.

Table 11: Additional hyperparameters for HGCBC, HGCBC-$\phi$, HBYOL-$\gamma$. For other hyperparameters we match those in Table 4. For high-level policies $\pi_h$ that predict in representation space (HGCBC-$\phi$, HBYOL-$\gamma$), we find it is better to use a smaller learning rate.

| Hyperparameter | Value |
|---|---|
| Hierarchical head | MLP (512, 512, 512, 512) |
| Low-level head | MLP (512, 512, 512) |
| Sub-goal steps | 25 |
| Learning rate | $3 \times 10^{-4}$ (HGCBC), $10^{-4}$ (HGCBC-$\phi$, HBYOL-$\gamma$) |

# D  CL TO TD-SR

Here, illustrate that connection between CL and TD-SR, showing that in the limit an n-step version of TD-SR becomes similar to CL.

We can rewrite Equation (1) to see the connection between TD-SR and CL (MC). Under assumptions that $f$ is the dot product between $\phi$ and $\psi$, and $\phi, \psi$ are centered, if we apply a second-order Taylor expansion to the denominator of the CL loss (Touati et al., 2023) we have:

$$\text{CL}_{\text{InfoNCE}} \approx \mathbb{E}_{s \sim p, s' \sim p} \left[ (\psi(s)^T \phi(s'))^2 \right] - 2 \mathbb{E}_{\substack{k \sim \text{geom}(1-\gamma) \\ s_t \sim p, s_{t+k} \sim p^\pi(s_{t+k}|s_t)}} \left[ \psi(s_t)^T \phi(s_{t+k}) \right] \quad (14)$$

Next, we can consider an n-step variant of the TD-SR loss (Blier et al., 2021) which we refer to as TD-SR($n$):

$$\min_{\phi, \psi} \mathbb{E}_{\substack{s_t \sim p \\ s' \sim p}} \left[ (\psi(s_t)^T \phi(s') - \gamma^n \bar{\psi}(s_{t+n})^T \bar{\phi}(s'))^2 \right] - 2 \sum_{i=1}^{n} \mathbb{E}_{s_t \sim p, s_{t+i} \sim p^\pi} \left[ \gamma^i \psi(s_t)^T \phi(s_{t+i}) \right]$$

$$(15)$$

We can make the full connection to CL with infinite horizon $n$:

$$\text{TD-SR}(n) = \mathbb{E}_{\substack{s_t \sim p \\ s' \sim p}} \left[ (\psi(s_t)^T \phi(s'))^2 \right] - 2 \sum_{i=1}^{n} \mathbb{E}_{\substack{s_t \sim p \\ s_{t+i} \sim p^\pi(s_{t+i} s_0)}} \left[ \gamma^i \psi(s_t)^T \phi(s_{t+i}) \right] \qquad (16)$$

$$= \mathbb{E}_{\substack{s_t \sim p \\ s' \sim p}} \left[ (\psi(s_t)^T \phi(s'))^2 \right] - \frac{2\gamma}{(1-\gamma)} \sum_{i=1}^{n} \mathbb{E}_{\substack{s_t \sim p \\ s_{t+i} \sim p^\pi(s_{t+i} s_0)}} \left[ (1-\gamma) \gamma^{i-1} \psi(s_t)^T \phi(s_{t+i}) \right] \qquad (17)$$

$$= \mathbb{E}_{\substack{s_t \sim p \\ s' \sim p}} \left[ (\psi(s_t)^T \phi(s'))^2 \right] - \frac{2\gamma}{(1-\gamma)} \mathbb{E}_{\substack{k \sim \text{geom}(1-\gamma) \\ s_t \sim p, s_{t+k} \sim p^\pi(s_{t+k}|s_t)}} \left[ \psi(s_t)^T \phi(s_{t+k}) \right] \qquad (18)$$

Thus, we can see that in the infinite horizon form of $\text{TD-SR}(n)$, it is related to the form of $\text{CL}_{\text{InfoNCE}}$ in (1), but with the positive contrastive term weighted by factor $\frac{\gamma}{1-\gamma}$.

## E  FINITE MDP

### E.1  BYOL

**BYOL as an Ordinary Differential Equation (ODE)**  In finite MDPs, we can characterize the BYOL objective which gives intuition about what information is captured in $\phi, \psi$, and conditions that may be useful for stability (Tang et al., 2023; Khetarpal et al., 2025). Consider a finite MDP with transition $P^\pi$, linear d-dimensional encoder $\Phi \in \mathbb{R}^{|S| \times d}$, and linear action-free latent-dynamics $\Psi \in \mathbb{R}^{d \times d}$. In a finite MDP, Equation (3) becomes:

$$\min_{\Phi, \Psi} \text{BYOL}(\Phi, \Psi) := \min_{\Phi, \Psi} \mathbb{E}_{s_t \sim p_0(s), s_{t+1} \sim P^\pi}, \left[ \| \psi^T \Phi^T s_t - \bar{\Phi}^T s_{t+1} \|_2^2 \right] \qquad (19)$$

A property to prevent this objective from collapsing is that $\Psi$ is updated more quickly than $\Phi$. In practice, this is commonly realized as the dynamics are generally a smaller network than the encoder. This system can be analyzed in an ideal setup, where we first find the optimal $\Psi$, each time before taking a gradient step for $\Phi$, which leads to the ODE for representations $\Phi$ (Tang et al., 2023):

$$\Psi^* \in \arg\min_{\Psi} \text{BYOL}(\Phi, \Psi), \quad \dot{\Phi} = -\nabla_\Phi \text{BYOL}(\Phi, \Psi)|_{\Psi = \Psi^*} \qquad (20)$$

We are able to analyze this ODE with the following assumptions (Tang et al., 2023):

**Assumption E.1** (Orthogonal initialization). $\Phi^\top \Phi = I$

**Assumption E.2** (Uniform state distribution). $p_0(s) = \frac{1}{|\mathcal{S}|}$

**Assumption E.3** (Symmetric dynamics). $P^\pi = (P^\pi)^\top$

Under these three assumptions, Khetarpal et al. (2025) prove that the BYOL ODE is equivalent to monotonically minimizing the surrogate objective:

$$\min_{\Psi} \| P^\pi - \Phi \Psi \Phi^T \|_F + C \qquad (21)$$

Where $\| \cdot \|_F$ is the Frobenius matrix norm. Thus, we can understand that the BYOL objective as learning a d-rank decomposition of the underlying dynamics $P^\pi$. Additionally, the top $d$ eigenvectors of $P^\pi$ match those of $(I - \gamma P^\pi)^{-1} = M^\pi$ (Chandak et al., 2023). However, we will highlight that there are key differences when *learning* a low-rank decomposition between $P^\pi$ and $M^\pi$. This is described by Touati et al. (2023), where we can consider that in a real-world problem with underlying continuous-time dynamics, actions may have little effect, and $P^\pi$ is close to the identity, i.e. close to full-rank. However, $M^\pi$, which takes powers of $(P^\pi)^t$, has a "sharpening effect" on the difference between eigenvalues, which gives a clearer learning signal. This is intuitive on a real-world problem like robotics, even with discrete-time dynamics, where $s_{t+1} \approx s_t$, but we have larger differences between $s_t$ and $s_{t+k}$.

### E.2  BYOL-$\gamma$

**In the finite MDP**, we now verify theorem 4.1 , where BYOL-$\gamma$ approximates the successor representation with matrix decomposition $\tilde{M}^\pi \approx \Phi \Psi \Phi^T$.

We consider the same objective (19), where we need to update the expectation of the sampling distribution:

$$\min_{\Phi,\Psi} \text{BYOL-}\gamma(\Phi,\Psi) := \min_{\Phi,\Psi} \mathbb{E}_{s_t \sim p_0(s), s_+ \sim \tilde{M}^\pi, } \left[ \|\psi^T \Phi^T s_t - \bar{\Phi}^T s_+\|_2^2 \right] \tag{22}$$

Assuming that this objective is optimized under the ODE (20). We have that our objective monotonically minimizes:

$$\min_{\Psi} \|\tilde{M}^\pi - \Phi\Psi\Phi^T\|_F + C \tag{23}$$

This directly translates as we can consider $\tilde{M}^\pi = P^\pi$ as simply a valid transition matrix for a new, temporally abstract, version of the original MDP. We maintain the original assumptions E.1, E.2, and E.3. We do not need an additional assumption for $\tilde{M}^\pi$, as assumption E.3 for symmetric $P^\pi$ implies a symmetric $\tilde{M}^\pi = (1-\gamma)\sum_{t\geq 0}\gamma^t P_\pi^t$,

Under this setup, we also have that $\Psi\Phi \in \mathbb{R}^{n\times d}$ relates to the successor feature matrix, where each row $(\Psi\Phi)_i$ contains the vector $(1-\gamma)\psi^\pi(s_i)$:

$$(1-\gamma)\psi^\pi(s_i) = \sum_j \tilde{M}^\pi(s_i, s_j)\phi(s_j) \tag{24}$$

$$= (\tilde{M}^\pi\Phi)_i \tag{25}$$

$$\approx (\Phi\Psi\Phi^T\Phi)_i \tag{26}$$

$$= (\Phi\Psi)_i \tag{27}$$

In other words, in the restricted finite MDP, where we minimize (23), we are simultaneously learning successor features $\psi^\pi \approx \Psi\Phi$ and basis features $\Phi$.

## F    MIXTURE DATASETS

In Section 3.2, we describe a practical setting where we have an offline dataset generated by a set of policies $\{\beta_j\}$. While we previously describe that BYOL-$\gamma$ approximates $\tilde{M}^\pi$ when we have MC samples directly an arbitrary $\pi$, we now describe the behavior of BYOL-$\gamma$ when trained jointly on MC samples from multiple $\{\beta_j\}$, first in the finite MDP, and how this relates to approximating the SR of the unknown mixture policy $\tilde{M}^\beta$.

**SR of Mixture Policy**.    We begin by obtaining the SR for the mixture policy $\beta(a|s) := \sum_j \beta_j(a|s)p(\beta_j|s)$, first defining 1-step transitions:

$$P_{i,l}^{\beta_j} = p^{\beta_j}(s_{t+1} = l|s_t = i) = \sum_a \beta_j(a|s = i)p(s_{t+1} = l \mid s_t = i, a) \tag{28}$$

$$P_{i,l}^\beta = \sum_a \sum_j \beta_j(a|s)p(\beta_j|s)p(s_{t+1} = l|s_t = i, a) = \sum_j p(\beta_j|s = i)P_{i,l}^{\beta_j} \tag{29}$$

Using $w_j(i) = p(\beta_j|s = i)$, $W_j = \text{diag}(w_j(1), \cdots, w_j(|S|))$, we can see the transitions of the mixture policy $\beta$ as simply a (state-dependent) weighted average of the transitions of $\{\beta_j\}$.

$$P^\beta = \sum_j W_j P^{\beta_j} \tag{30}$$

$$\tilde{M}^\beta = (1-\gamma)\sum_{t\geq 0}\gamma^{t+1}(\sum_j W_j P^{\beta_j})^{t+1} \tag{31}$$

**Approximated SR of BYOL-$\gamma$**. Using samples from a set of unknown policies $\{\beta_j\}$ the BYOL-$\gamma$ objective corresponds to:

$$\min_{\Phi,\Psi} \mathbb{E}_{s_t \sim p(s), \beta_j \sim p(\beta_j|s_t), s_+ \sim \tilde{M}^{\beta_j}} \left[ \|\psi^T\Phi^T s_t - \bar{\Phi}^T s_+\|_2^2 \right] \tag{32}$$

$$= \min_{\Phi,\Psi} \mathbb{E}_{s_t \sim p(s), s_+ \sim \widehat{M}} \left[ \|\psi^T\Phi^T s_t - \bar{\Phi}^T s_+\|_2^2 \right] \tag{33}$$

i.e. by theorem 4.1 we are approximating $\widehat{M} = \sum_j p(\beta_j|s_t)\tilde{M}^{\beta_j}$, which we can compare to Equation (31) via:

$$\widehat{M} = (1 - \gamma)\sum_{t \geq 0}\gamma^{t+1}\sum_j W_j(P^{\beta_j})^{t+1} \tag{34}$$

Intuitively, while $\tilde{M}^\beta$ corresponds to the *SR of the average policy*, BYOL-$\gamma$ approximates $\widehat{M}$, *an average of policy SRs*.

**General Case**. We rewrite the inequality from Equation (9) but with mixture data:

$$\mathcal{L}_{\text{BYOL-}\gamma}(\phi, \psi) = \mathbb{E}_{\beta_j \sim p(\beta_j), s_t \sim p^{\beta_j}(s), s_+ \sim \tilde{M}^{\beta_j}(s_t, s_+)}\left[f(\psi(\phi(s_t)), \bar{\phi}(s_+))\right] \tag{35}$$

$$\geq \mathbb{E}_{\beta_j \sim p(\beta_j), s_t \sim p^{\beta_j}(s)}\left[f(\psi(\phi(s_t)), \mathbb{E}_{s_+ \sim \tilde{M}^{\beta_j}(s_t, s_+)}\bar{\phi}(s_+))\right]$$

$$= \mathbb{E}_{\beta_j \sim p(\beta_j), s_t \sim p^{\beta_j}(s)}\left[f(\psi(\phi(s_t)), (1 - \gamma)\psi_{\bar{\phi}}^{\beta_j}(s_t))\right]$$

### F.1 CL on Mixture Data

We discuss the behavior of CL on mixture datasets. First, we write Equation (36), we rewrite Equation (1) when practically applied to mixture data as implemented in TRA:

$$\mathcal{L}_{\text{TRA}} \approx \mathbb{E}_{\substack{\beta_j \sim p(\beta_j), s_t \sim p_j^\beta(s) \\ s_+ \sim \tilde{M}^{\beta_j}(s_t, s_+)}}\left[f(\psi(s_t), \phi(s_+))\right] - \mathbb{E}_{s^{1:N} \sim p_j^\beta(s)}\left[\log\sum_{i=2}^N e^{f(\psi(s^1), \phi(s^i))}\right] \tag{36}$$

We note that a mismatch occurs between the numerator (attractive), and the denominator (repulsive) terms. Namely, we attract two representations only when they are sampled from the same policy $\beta_j$, but minimize similarly for states sampled under the occupancy of the mixture policy $\beta$.

We could consider other forms where both terms sample from the same distributions. Namely, in the ideal case if we could get MC samples from $s_+ \sim \tilde{M}^\beta(s, s_+)$, the loss has the form:

$$\mathcal{L}_{\text{CL}^\beta} \approx \mathbb{E}_{\substack{s_t \sim p^\beta(s) \\ s_+ \sim \tilde{M}^\beta(s_t, s_+)}}\left[f(\psi(s_t), \phi(s_+))\right] - \mathbb{E}_{s^{1:N} \sim p^\beta(s)}\left[\log\sum_{i=2}^N e^{f(\psi(s^1), \phi(s^i))}\right] \tag{37}$$

In practice, we only can take MC samples from, $\beta_j$'s, so we could view the loss as an expectation over these policies:

$$\mathcal{L}_{\text{CL}^{\beta_j}} \approx \mathbb{E}_{\substack{\beta_j \sim p(\beta_j), s_t \sim p^{\beta_j}(s) \\ s_+ \sim \tilde{M}^{\beta_j}(s_t, s_+)}}\left[f(\psi(s_t), \phi(s_+))\right] - \mathbb{E}_{\beta_j \sim p(\beta_j), s^{1:N} \sim p^{\beta_j}(s)}\left[\log\sum_{i=2}^N e^{f(\psi(s^1), \phi(s^i))}\right]$$

$$= \mathbb{E}_{\beta_j}\left[\mathbb{E}_{\substack{s_t \sim p^{\beta_j}(s) \\ s_+ \sim \tilde{M}^{\beta_j}(s_t, s_+)}}\left[f(\psi(s_t), \phi(s_+))\right] - \mathbb{E}_{s^{1:N} \sim p^{\beta_j}(s)}\left[\log\sum_{i=2}^N e^{f(\psi(s^1), \phi(s^i))}\right]\right] \tag{38}$$

This objective similarly corresponds to capturing information related to a mixture of different policies, similarly to the BYOL-$\gamma$ objective. We can see the positive term of $\mathcal{L}_{\text{TRA}}$ matches $\mathcal{L}_{\text{CL}^{\beta_j}}$ while the negative term matches $\mathcal{L}_{\text{CL}^\beta}$. In other words, $\mathcal{L}_{\text{TRA}}$ is an under-optimistic compared to $\mathcal{L}_{\text{CL}^\beta}$ and over-pessimistic compared to $\mathcal{L}_{\text{CL}^{\beta_j}}$. We can see how this may discourage stitching. For example, if we have a trajectory $a \to b$, $b \to c$. Although we want relation from $a, c$, $\psi(a)\phi(c)$ is only sampled as a negative term.

**SVD approximation of TRA.** In the single policy case, Touati et al. (2023) demonstrates that CL with a single policy ($\mathcal{L}_{\text{CL}}\beta$) relates to an SVD of $\frac{\tilde{M}^\beta(s,s')}{p^\beta(s')}$:

$$\mathcal{L}_{\text{CL}}\beta \approx \mathbb{E}_{s \sim p^\beta, s' \sim p^\beta}\left[\left(\frac{\tilde{M}^\beta(s, s')}{p^\beta(s')} - \psi(s)^T\phi(s')\right)^2\right] + C$$

We now show that the mixture policy case corresponds to an SVD of $\frac{\sum_j p(\beta_j|s)\tilde{M}^{\beta_j}(s,s_+)}{p^\beta(s_+)}$:

$$
\begin{aligned}
\mathcal{L}_{\text{TRA}} &\approx \mathbb{E}_{\substack{\beta_j \sim p(\beta_j), s_t \sim p^{\beta_j} \\ s' \sim \tilde{M}^{\beta_j}(s_t, s')}} \left[ f(\psi(s_t), \phi(s')) \right] - \mathbb{E}_{s \sim p^\beta} \left[ \log \mathbb{E}_{s' \sim p^\beta} [e^{f(\psi(s), \phi(s'))}] \right] \\
&= \mathbb{E}_{s \sim p^\beta, s' \sim p^\beta} \left[ \frac{\sum_j p(\beta_j|s)\tilde{M}^{\beta_j}(s, s')}{p^\beta(s')} f(\psi(s), \phi(s)) \right] - \mathbb{E}_{s \sim p^\beta} \left[ \log \mathbb{E}_{s' \sim p^\beta} [e^{f(\psi(s), \phi(s'))}] \right]
\end{aligned}
$$
(39)

Under assumptions that $f$ is the dot product between $\phi$ and $\psi$, and $\phi, \psi$ are centered, if we apply a second-order Taylor expansion to second term:

$$
= \mathbb{E}_{s \sim p^\beta, s' \sim p^\beta} \left[ \frac{\sum_j p(\beta_j|s)\tilde{M}^{\beta_j}(s, s')}{p^\beta(s')} \psi(s)^T \phi(s') \right] - \frac{1}{2} \mathbb{E}_{s \sim p^\beta, s' \sim p^\beta} \left[ (\psi(s)^T \phi(s'))^2 \right]
$$
(40)

$$
= \mathbb{E}_{s \sim p^\beta, s' \sim p^\beta} \left[ \left( \frac{\sum_j p(\beta_j|s)\tilde{M}^{\beta_j}(s, s')}{p^\beta(s')} - \psi(s)^T \phi(s') \right)^2 \right]
$$
(41)

# G    ADDITIONAL RESULTS FOR HORIZON GENERALIZATION

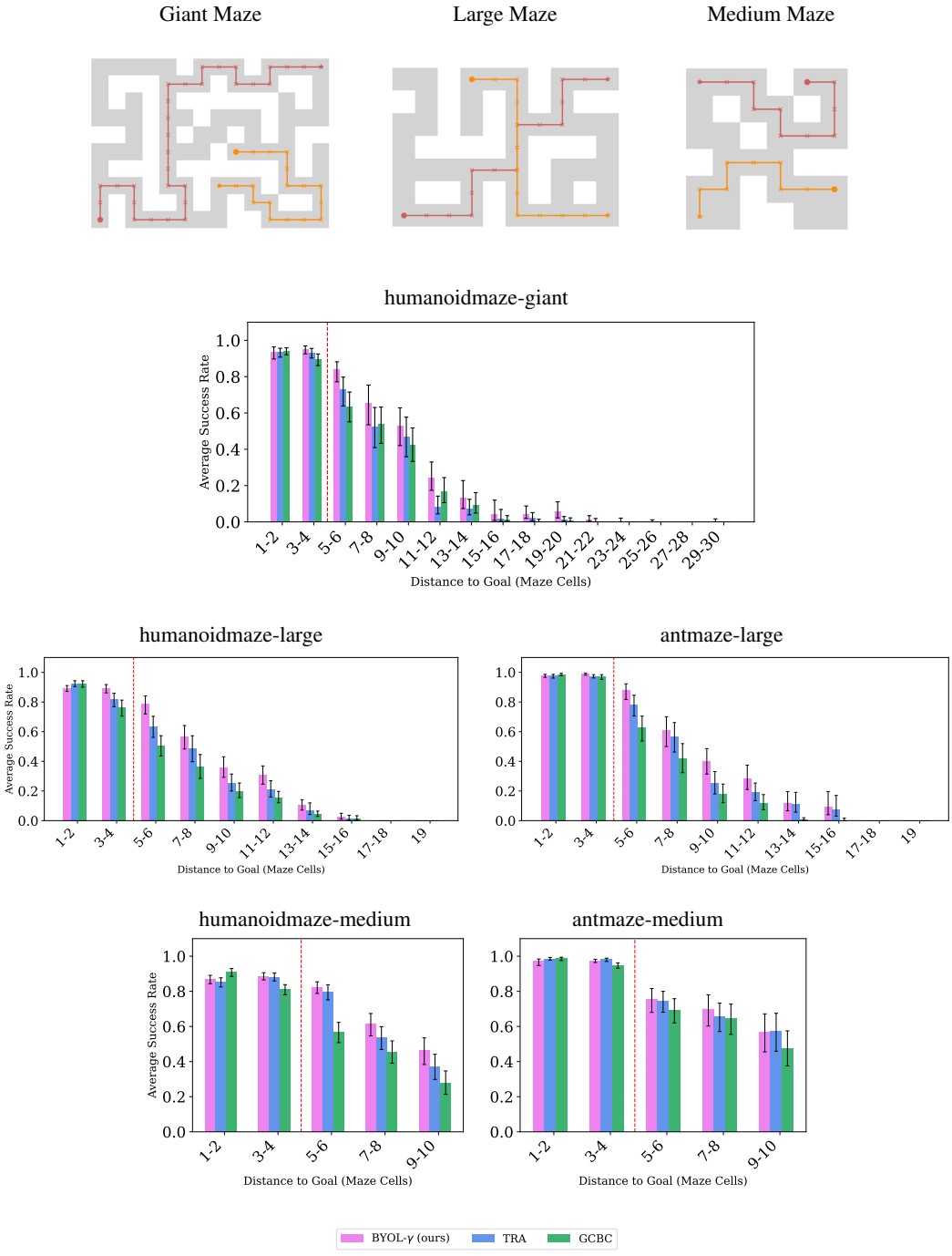

Figure 6: **Evaluating Generalization with Increasing Horizons:** The distances to the right of the red dotted line require combinatorial generalization. The maze maps show examples of how intermediate goals are selected along the optimal path.

We include additional results matching the setup in Section 5.3, for `antmaze-medium`, and `{humanoidmaze}-{medium,large,giant}` in Figure 6. We can observe that BYOL-$\gamma$ leads in performance as the distance between the start and goal grows when compared to other methods.

# H  REPRESENTATIONS

## H.1  ADDITIONAL VISUALIZATIONS

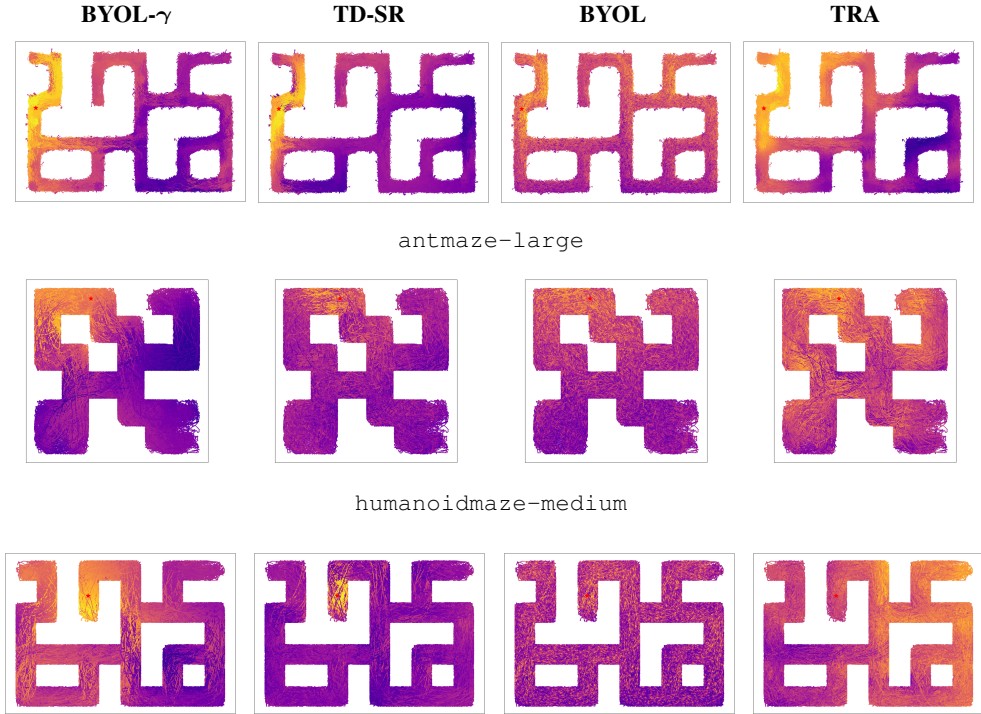

Figure 7: **Additional Visualization of the Learned Representation**: depicts the similarity between the prediction of the current state representation to the goal representation. Brighter color indicates higher similarity.

## H.2  CORRELATION TO SHORTEST PATH

We conduct a quantitative comparison between representations through alignment with shortest-path distance in the environment. Namely, we compute the correlation between similarity in the representation space, $\frac{\psi(s,\cdot)^T \phi(g)}{\|\psi(s,\cdot)\|\|\phi(g)\|}$, to the shortest path distance in the maze between sampled start and goal cells ($xy$ space). While the shortest path distance does not measure ground truth temporal distance, as it does not account for robot dynamics, it still provides a simple reference for the general structure we expect to see in representations. We can see that in Table 12, on average BYOL-$\gamma$'s representations seem to most strongly correlate with shortest path distance. We also compute the success rate of the same checkpoints used for correlation, and notice relationships between these correlations and empirical success rate. We see that the ranking of methods in terms of average correlation in representation space matches the ordering of methods in terms of average empirical policy success.

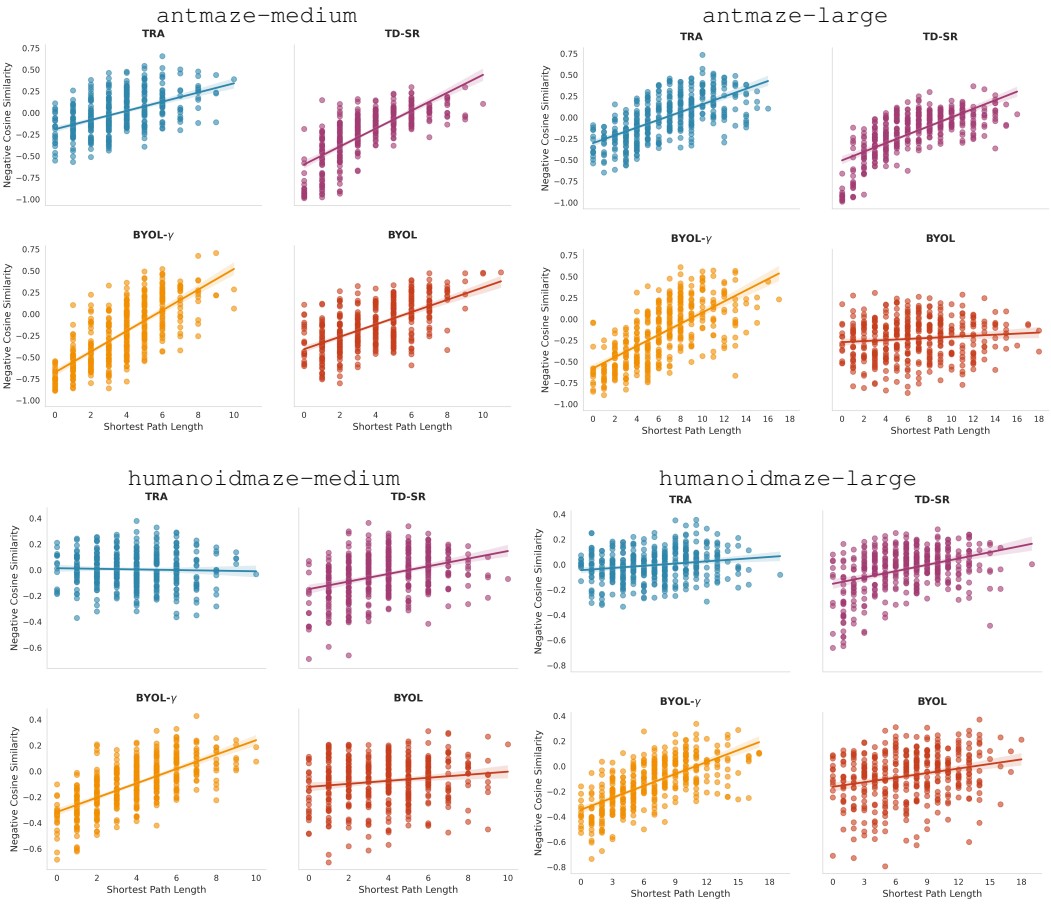

Figure 8: **Scatter plot of negative cosine similarity between randomly sampled (state,goal) pairs in representation spaces and true shortest path**, aggregated over 4 model seeds, each sampled at 100 (state,goal) pairs.

| Dataset | BYOL-$\gamma^a$ | BYOL | TRA | TDSR$^a$ |
|---|---|---|---|---|
| antmaze-medium-stitch | $0.71 \pm 0.01$ | $0.59 \pm 0.05$ | $0.49 \pm 0.05$ | $0.72 \pm 0.03$ |
| antmaze-large-stitch | $0.66 \pm 0.02$ | $0.10 \pm 0.02$ | $0.62 \pm 0.03$ | $0.67 \pm 0.02$ |
| humanoidmaze-medium-stitch | $0.64 \pm 0.02$ | $0.18 \pm 0.04$ | $0.02 \pm 0.02$ | $0.36 \pm 0.03$ |
| humanoidmaze-large-stitch | $0.62 \pm 0.03$ | $0.20 \pm 0.03$ | $0.15 \pm 0.04$ | $0.38 \pm 0.03$ |
| average maze correlation | **0.66** | 0.27 | 0.32 | 0.53 |
| average maze success | **39** | 26 | 31 | 36 |

Table 12: **Correlation of representation space with shortest path distance**. For each method, we use $10,000$ (state, goal) pairs to compute correlation, and then compute the average and standard deviation of the correlation over 4 model seeds, and the success rate over these same checkpoints.

