# OpenReview forum: "Self-Predictive Representations for Combinatorial Generalization in Behavioral Cloning"
_ICLR.cc/2026/Conference — ICLR 2026 Poster_

### Official Review · Reviewer_c6Ng · 2025-10-27

**Soundness:** 2
**Presentation:** 2
**Contribution:** 2
**Rating:** 4
**Confidence:** 4

**Summary:**

This paper proposed BYOL-$\gamma$ for learning representations in goal-conditioned reinforcement learning (GCRL) problems. The core idea of BYOL-$\gamma$ is simple: instead of learning representations by aligning one-step latent transitions as in prior RL methods utilizing BYOL, BYOL-$\gamma$ learns representations by aligning multi-step latent transitions following the successor measures. Empirically, BYOL-$\gamma$ achieves competitive empirical performance across a suite of challenging tasks from the OGBench benchmarks.

**Strengths:**

- Learning representations for control is an important topic in RL. The paper motivates learning temporal representation for the GCBC problem by using the successor measure. Instead of bootstrapping the representations of the next state as in BYOL, BYOL-$\gamma$ proposes to bootstrap the representations of a future state sampled from the successor measure, which encodes temporal information in the dataset. The learned representations are then used to drive the learning of the goal-conditioned policy via behavioral cloning.

- The author conducted experiments on the challenging OGBench benchmarks, showing that BYOL-$\gamma$ achieves competitive empirical performance, compared to prior representation learning methods for GCBC and prior goal-conditioned reinforcement learning methods in the offline setting. Additional experiments ablate the key components of BYOL-$\gamma$ and demonstrate that BYOL-$\gamma$ can generalize over longer horizons than prior methods.

**Weaknesses:**

-  The paper reviewed different representation learning objectives for RL in Sec. 3.1 and reviewed the generalization gap from prior work in Sec. 3.2 and the beginning of Sec. 4 (line 223 - line 249). However, Sec. 4.1 proposed the BYOL-$\gamma$ objective directly without explaining its connection with the preliminaries. It is difficult to understand the roles of Sec. 3.1, Sec. 3.2, and the beginning of Sec. 4 in the paper.

- One of the motivations of developing BYOL-$\gamma$ is to learn representations that enable combinatorial generalization in GCRL. The objective function in Eq. 8, the following Eq. 9 and Theorem 4.1 do not provide enough explanations for why the objective function can enable combinatorial generalization.

- Sec 4.2 introduces another auxiliary loss function called TD-SR for learning representations. The relationships between TD-SR and BYOL-$\gamma$ are not clear from the current draft, making it unclear whether the paper proposed BYOL-$\gamma$ or TD-SR as its main representation learning objective.

- Results in Table 3 seem to suggest some components of BYOL-$\gamma$ has minor effects.

Overall, the motivations, the algorithms, and the theory are somewhat inconsistent.

**Questions:**

- In Sec. 3.1, what are the connections between CL and TD-SR to BYOL-$\gamma$? Does the connection between CL and TD-SR, line 174: "an n-step version of TD-SR is related to CL", help us to understand BYOL-$\gamma$?

- Sec. 3.2 and the beginning of Sec. 4 reviewed prior definition of generalization gap in GCBC. What are the connections between these definitions and the BYOL-$\gamma$ objective in Sec. 4.1?

- The motivation of developing the objective in Eq. 8 is to enable combinatorial generalization. But why does this objective enable combinatorial generalization? What is the policy $\pi$ in this equation? Is it the behavioral policy or the learned policy?

- Eq. 10 includes a bidirectional prediction term. Any explanation for the reason for including this bidirectional prediction term? Results in Table 3 seems to suggest the effect of this bidirectional prediction term is minor.

- Sec 4.2 introduces another auxiliary loss function called TD-SR. What’s the relationship between TD-SR and BYOL-$\gamma$? Does the paper propose TD-SR or BYOL-$\gamma$ as the main representation learning objective?

- The second and third columns in Table 1 are confusing. From line 205, it looks like $\tilde{M}^{\beta}(s, s_{+})$ and $\sum_j p(\beta_j \mid s) \tilde{M}^{\beta_j}(s, s_{+})$ are equivalent to each other. What’s the difference between these two columns in the context of different methods?

- Line 335: More loss terms with the same batch of dataset typically results in higher learning efficiency. Is this sentence saying that BYOL-$\gamma$ is less efficient than CL and TD-SR?

- Sec 5.1 visualizes the representation learned by different methods. From Fig 2, it looks like the cosine similarity of BYOL-$\gamma$ and TD-SR are more concentrated than BYOL and TRA. How can we relate these representation heatmaps to the reachability of different states?

---

> ### Author Response · Authors · 2025-11-22
>
> Thank you for your thoughtful review of our paper and feedback. We are glad that your found our experiments on the challenging OGBench datasets to be interesting. We will address each of your concerns in turn below:
>
> __Q1: “proposed the BYOL-$\gamma$ objective directly without explaining its connection with the preliminaries”__
>
> A1: We apologize for any confusion created in our presentation of our method. Sections 3.1 and 3.2 present relevant background for our method, 3.1 introduces existing methods for representation learning that have connections to the successor measure while 3.2. presents existing results for the generalization gap in combinatorial generalization. The beginning of Sec 4, prior to 4.1 draws the connections together by presenting a theoretical intuition for why successor measure representation learning methods (sec 3.1) should help with generalization gap (3.2). This provides the context for the gap with a standard BYOL objective, and leads us to our method in 4.1, which is a new method for successor measure representation learning. In section 4.2 and Table 1, we bring these objectives together, highlighting concerns with prior representation learning (TRA), and demonstrates the benefit to applying  BYOL-$\gamma$ and TD-SR.
>
> __Q2: “Theorem 4.1 do not provide enough explanations for why the objective function can enable combinatorial generalization.”__
>
> A2: In Section 4, we provide a theoretical intuition for why learning representations with temporal consistency under the successor measure would be beneficial for stitching. In short, we wish to stitch trajectories from $s$ to $s_f$ through a waypoint $s_w$. By encoding $\phi(s_w)$ and $\phi(s_f)$ or more generally $\phi(s_f)$ where $s_f \sim M(s_w, s_f)$ close in representation space, we can achieve this stitching property purely through representations alone.Then, the subsequent sections (4.1 and 4.2) explain how to learn representations that achieve the property identified in equation 7.
>
> __Q3: “The relationships between TD-SR and BYOL-$\gamma$ are not clear from the current draft”__
>
> A3:. We present the TD-SR method here as our work is specifically exploring SM-representations in the context of GCBC auxiliary losses. While TD-SR is known to approximate the SM and has been explored in related fields such as zero-shot RL, our work is the first to identify that this objective is suited for inducing generalization in BC which is a very different setting from prior work.
>
> The primary connection between the methods is that both are approximations of the successor measure of different quantities which is defined in Table 1. Intuitively, TD-SR bootstraps its estimation of the successor measure with TD-learning while BYOL-$\gamma$ bootstraps in representation space by predicting future MC representations .
>
> As we show in Table 1 and discussed further in Appendix E, in datasets created by a mixture of policies, MC and TD methods approximate slightly different successor measures, with MC methods approximating a mixture of the SMs for each policy while TD methods approximate the SM of the mixture. We introduce BYOL-$\gamma$ as the primary contribution of our work, as a non-contrastive alternative to TRA, which we show achieves competitive results. While we apply our objective to the setting of improving BC, it is a new variant of the BYOL objective which may be  useful in other settings, such as for exploration, zero-shot RL, or GCRL.
>
> __Q4: “some components of BYOL- has minor effects”__
>
> A4: We agree that some components have minor affects and we include all ablations of components to be transparent. However, since each component confers some benefit, we felt it beneficial to include each component in our final method. As you suggest, however, we believe that our contribution of BYOl-$\gamma$ without any additional components constitutes the core contribution of our work.
>
> __Q5: Connections between TD-SR, CL and BYOL-$\gamma$__
>
> A5: These methods are connected in that they are all related to approximating the successor measure. We can think of CL as a Monte Carlo version of TD-SR, hence, much like regular TD learning converges to MC in the limit as n-step goes to infinity, so too can we think of CL converging to TD-SR. Both methods are contrastive, in the sense that they require negative samples. In our work, we introduce BYOL-$\gamma$, which we show also approximates the successor measure but is a _non-contrastive_ method. We elaborate further on the connections between these methods in Appendices D and E, where we prove Theorem 1, that BYOl-$\gamma$ approximates successor representations in the finite MDP setting and elaborate on the connections between CL and TD-SR.
>
> __Q6: Connections between sections 3.2, 4.1 and BYOL-$\gamma$__
>
> A6: Please see A1 above.
>
> __Q7: “But why does this objective enable combinatorial generalization”__
>
> A7: Please see A2 above.
>
> ---- continue in following comment -----

---

> > ### Author Response · Authors · 2025-11-22
> >
> > __Q8: “Any explanation for the reason for including this bidirectional prediction term?”__
> >
> > A8: Bidirectional prediction has nice theoretical properties in the one-step BYOL case as described in [1], namely that with bidirectional prediction a milder condition on the transition dynamics is required to have BYOL learn SVD decomposition of the transition matrix. Though we only extend their theory to the SM in our paper for the one-directional case, there is good theoretical motivation to look also at bidirectional prediction.
> >
> > __Q9: “What’s the relationship between TD-SR and BYOL-$\gamma$”__
> >
> > A9: Please see A3 above.
> >
> > __Q10: “The second and third columns in Table 1 are confusing”__
> >
> > A10: This Table is highlighting differences in the metric learned by various SM learning methods under the common GCBC setting of mixed-policy datasets. The two terms you reference are not equivalent in this setting, where the first term is the successor measure of the mixture policy, while the second term is a mixture of the successor measures of each individual policy. This is a subtle difference that motivates the use of TD methods like TD-SR for representation learning on mixed datasets, as it can better capture this mixture distribution. However, TD-SR relies on contrastive samples while our method is able to achieve similar representations to TD-SR without the use of contrastive samples.
> >
> > __Q11: “More loss terms with the same batch of dataset typically results”__
> >
> > A11: The sentence you are referring to is highlighting the compute efficiency of using non-contrastive methods like BYOL-$\gamma$ vs contrastive methods like TD-SR and CL, which require computing loss terms across pairs of states.
> >
> > __Q12: “relating these representation heatmaps to the reachability of different states”__
> >
> > A12:Please see our meta response for an evaluation of the correlation between representation similarity and shortest paths between states.
> >
> > [1] Tang, Yunhao, et al. "Understanding self-predictive learning for reinforcement learning." International Conference on Machine Learning. PMLR, 2023.

---

> ### Comment · Reviewer_c6Ng · 2025-11-27
>
> I thank the authors for their responses to my questions. I would like to mention three follow-ups:
>
> > In Section 4, we provide a theoretical intuition for why learning representations with temporal consistency under the successor measure would be beneficial for stitching.
>
> Even though the intuition is correct, I still find it difficult to understand why the BYOL-$\gamma$ objective (Eq. 10) follows this intuition. Specifically, what's the policy $\pi$ in Eq. 10? Is it the behavioral policy or the BC policy? I believe that clarifying the connection between the practical objective and the theoretical intuition is important to undertanding the core idea of the method.
>
> > The relationships between TD-SR and BYOL-$\gamma$
>
> If I understand correctly, BYOL-$\gamma$ is an MC method, and TD-SR is a TD method. In general, an MC method does **not** have the capability to do stitching (or combinatorial generalization in this paper), while a TD method has the capability because of the dynamic programming loss, e.g., Eq. 2.
>
> This paper tries to demonstrate the capability of stitching in BC. So if the final representation learning objective (line 311) is a combination of MC and TD loss. I would expect the stitching capability to come from the TD loss instead of the MC loss. In this case, does the BYOL-$\gamma$ itself have the stitching capability? And which algorithm is the core idea of this paper? BYOL-$\gamma$ or TD-SR
>
> > where the first term is the successor measure of the mixture policy, while the second term is a mixture of the successor measures of each individual policy
>
> According to the definition between lines 205 - 209, these two quantities are the same. Is this saying that Table 1 refers to other quantities? Can you explain the difference between these two quantities using the definitions in the paper?

---

> ### Author Response · Authors · 2025-12-02
>
> __Q2__ continued: understanding why the BYOL-objective (Eq. 10) enables combinatorial generalization, “what's the policy $\pi$  in Eq. 10?”
>
> Thanks for your questions, the source of confusion is more clear, and it may stem from the notation that $\pi$ is essentially used in two different scenarios.
>
> First, we $\pi$ is used as a generic policy symbol, and given we have direct samples from $\pi$ to define representation learning methods, which is sections 3.1 and then in 4.1 when we introduce BYOL-$\gamma$. Second, the notation for $\pi$ in Section 3.2., and 4.2, $\pi$ does not mean an arbitrary policy, but refers to the BC policy. To make this more clear, we have updated the label of the BC policy to $\pi_\Theta$ instead of $\pi$ in Sections. Thus, the notation in equation 10, $pi$, is a data collecting (behavioral) policy, not the BC policy.
>
>
> __Q3__ continued: relationship between TD-SR and BYOL-$\gamma$
>
> We would like to clarify how MC and TD based __representation learning__ methods behave. Namely, the statement “an MC method does not have the capability to do stitching” is more applicable to TD-based value estimation rather than what we study in this paper, representation learning. The behavior of MC,TD for temporal representation learning does not directly line up with the behavior of MC,TD value estimation.
>
> Let’s build an illustrating example. We have two trajectories, that are each only one transition long. $\tau_1 = n(s_j, r_j, s_k)$, $\tau_2 = (s_k, r_k, s_l)$. Assume $s_l$ is terminal. The notation reads that in the first trajectory,  we transition from $s_j$ to $s_k$ and get rewarded $r_j$. We can see that TD value estimation stitches, e.g. would estimate $V(s_j) = r_j + \gamma V(s_k) = r_j + \gamma r_k$ , but MC would just give $V(s_j) = r_j$,  not stitching.
>
> The situation is different in MC sampling for representation learning - with BYOL$\gamma$ we bootstrap the current representations with future representations, implicitly stitching trajectories. Consider our previous example - under BYOL-$\gamma$, we train state representation $\phi$ and latent predictor $psi$. From $\phi(s_j)$, you try to predict $\phi(s_k)$, i.e. $\psi(\phi(s_j))$ tries to update $\phi, $\psi to predict $\phi(s_k)$ in the first trajectory. However, in the second trajectory $\phi(s_k)$ tries to predict $\psi$  $\phi(s_l)$ via $\psi$, thus you can see that representations are related through, $\phi(s_j)$ approx → $\phi(s_k)$ approx →  $\phi(s_l)$, thus even with only MC sampling , and $s_j$ and $s_l$ are in different trajectories, we get that $\phi(s_j)$ is related to $\phi(s_l)$. Thus, from this example we can intuitively see how BYOL-\gamma leads to stitching in representation space.
>
> The intuition is that because you get this stitching effect, just by predicting future representations (of states sampled MC), it may not be necessary to do TD in representation space (as in TDSR). Empirically, our paper demonstrates that this intuition holds, as with MC representation prediction, we outperform other methods and learn representations that capture stronger temporal structure on average than TD-SR (Appendix H, Table 12).
>
> __Q10__ continued:  “The second and third columns in Table 1 are confusing”, “...these two quantities are the same”
>
> Lines 205-209 (211 - 213 in current revision) do not imply that these columns are the same. First, in Section 3 we clarify that $M^\pi(s') = {E}_{s \sim p_0(s)}[M^\pi(s_0, s')]$ refers to the state occupancy. Lines 211 - 213  say that M^\beta(s), the state occupancy of the mixture policy, matches the expected state occupancy under the individual policies $\beta_j$. However, their conditional occupancies (the successor measures), do not match, $M^{\beta}(s, s’) \neq$ $E^{\beta j}[M^{\beta j}(s,s’)]$, which leads to the generalization gap in Equation 5.
>
> In regards to Table 1, the first column lists the objects that represent learning methods approximate in finite MDPs when we have direct trajectory samples from the mixture policy, while the second column describes behavior when we have trajectory samples from a mixture of separate policies. This gives intuition that we may expect to see a gap between MC and TD-based methods, and also on the differences between BYOL-$\gamma$ and contrastive learning.

---

### Official Review · Reviewer_JKTQ · 2025-10-29

**Soundness:** 3
**Presentation:** 3
**Contribution:** 2
**Rating:** 6
**Confidence:** 3

**Summary:**

This paper investigates the question of representation learning to improve combinatorial generalization in behavior cloning: what learning objective best recovers the latent dynamics of the environment? They focus on the self-predictive Bring Your Own Latent framework, introducing a modified objective BYOL-$\gamma$ that displays better ability to stitch together long trajectories, enabling better generalization in longer-horizon tasks.

**Strengths:**

1. The writing of the paper is generally good (easy to follow, good motivation of the problem, good coverage of related work)
    - In particular, the paragraph that discusses successor representations at the beginning of section 3 was very clear
2. The introduction to the experiments section (section 5 on page 7) is very nice and thorough
3. The various kinds of experimental results reported (eg, Figures 2 and 3) are interesting
    - The decision to "bold values within 95% of [the best] value in the same row" (table 2) is a great idea, very helpful for quickly parsing/comparing methods within each row
    - The latent state representation visualizations are very interesting
4. The inclusion of a limitations section at the end of the paper is appreciated
5. The BYOL-$\gamma$ method is motivated both intuitively and theoretically
    - Because of this, as well as because the paper seeks to study the broader topic of representation-learning objectives, I would say that the simplicity of the method (i.e., the similarity to the prior BYOL objective, eq. 3) is not really a weakness.

**Weaknesses:**

1. The theoretical assumption of symmetric transition dynamics does not seem realistic.
2. The qualitative results of the representation visualizations aren't convincing: the representation similarities for BYOL-$\gamma$ do not look clearly better than, e.g., BYOL and TRA.
    - In mazes, presumably (based on the concepts in this paper) the embedding similarity should decrease roughly monotonically with distance from the goal; thus, along with the provided visuals in Figures 2 and 6, it could be helpful to provide a one-dimensional scatterplot of "similarity (e.g., dot product) vs. true shortest-path distance from goal" for each sampled point in the space; this could facilitate easier (and less subjective) comparison of the methods.
3. The results in Figure 5 (Appendix F) seem to show that BYOL-$\gamma$ is not really better than the other methods (i.e., the plot for Figure 3 in the main text seems cherry-picked).
4. It could be useful to study some more interesting environments of different types, since all the experiments are just done in various mazes.

**Questions:**

1. The topic I am most unclear on is that equations (7) and (8) seem that they would be prone to collapse. How is this avoided? Is it related to the use of `stop gradient`? (the below points should be interpreted as both parts of this question and notes/suggestions)
    - The invariance $\phi(s_f) \approx \phi(s_w)$ seems that, without any "counter-balancing" from a term that differentiates states, it would coerce $\phi(\cdot)$ to encode all states similarly. How is this avoided?
    - The theoretical results in Appendix D mention stability, but it could be helpful to also discuss this in the main text.
2. This is not a question, but I wanted to note that the referenced paper (Richens et al., 2025) reminded me of the paper "Resolving Causal Confusion in Reinforcement Learning via Robust Exploration" (Lyle et al., 2021) that similarly discusses the importance of learning about the environment in order to generalize. This is not to say that it should be cited here; I just thought it might be interesting to the authors.

---

> ### Author Response · Authors · 2025-11-22
>
> We appreciate your thoughtful review of our paper and feedback.
>
> __Q1:“The theoretical assumption of symmetric transition dynamics“__
>
> A1: We note that the assumption of symmetric dynamics is used for the analysis in the BYOL ODE [1,2], however BYOL objectives have been empirically effective in environments in which this does not hold. We also note that the bidirectional prediction objective [1] was proposed to help improve the stability of BYOL, and uses the weaker assumption that $(P^\pi)^T$ is a valid transition matrix rather than fully symmetric which we apply to our setting.
>
> __Q2: “The results in Figure 5 (Appendix F) seem to show that BYOL-$\gamma$- is not really better than the other methods i.e., the plot for Figure 3 in the main text seems cherry-picked”__
>
> A2: We do not believe that the heatmaps in the main paper seem significantly better for BYOL-$\gamma$ compared to those in the Appendix. We expect to see states closer to the goal (represented as a red dot in the heat map) to be more similar (yellow), and decrease to darker color as you go further away.  We believe the heatmaps in the Appendix display this property most clearly for BYOL-$\gamma$ and TD-SR. For example, in antmaze-large, the goal is in the top-left arm of the maze. In temporal space, this should be far from the middle-arm of the maze since reaching this arm would involve traversing along the middle corridor and up to the middle arm’s end. Both BYOL-$\gamma$ and TD-SR capture this with low similarity (purple) but both BYOL and TRA have high similarity (yellow).
>
> Thank you for suggesting the shortest path comparison. We have computed the suggested similarity metric and included the results in the general response as well as Appendix H.2. We also provide them here for your convenience.
>
>  | Dataset                    | BYOL-γ      | BYOL        | TRA         | TD-SR        |
> |----------------------------|-------------|-------------|-------------|-------------|
> | antmaze-medium-stitch      | 0.71 ± 0.01 | 0.59 ± 0.05 | 0.49 ± 0.05 | 0.72 ± 0.03 |
> | antmaze-large-stitch       | 0.66 ± 0.02 | 0.10 ± 0.02 | 0.62 ± 0.03 | 0.67 ± 0.02 |
> | humanoidmaze-medium-stitch | 0.64 ± 0.02 | 0.18 ± 0.04 | 0.02 ± 0.02 | 0.36 ± 0.03 |
> | humanoidmaze-large-stitch  | 0.62 ± 0.03 | 0.20 ± 0.03 | 0.15 ± 0.04 | 0.38 ± 0.03 |
> | average                    | 0.66        | 0.27        | 0.32        | 0.53        |
>
> We can see that on average,  BYOL-$\gamma$’s representations best capture shortest path distance.
>
> __Q3: “they would be prone to collapse. How is this avoided?”__
>
> A3: In general, representation collapse can be an issue for self-predictive methods like BYOL [3]. The general strategy for avoiding this collapse is to use a stop gradient and employ an exponential moving average on the target, which has been successful at preventing collapse in other settings, including RL [4][1]. Additionally, we note that for our method, which uses self-predictive representations as an auxiliary loss for BC, the BC loss itself acts as a strong deterrence against representation collapse.The representations passed to the policy are not frozen, rather the  BC loss is allowed to back-propagate through the representations. This encourages representations which are meaningful for predicting expert actions.
>
> [1] Tang, Yunhao, et al. "Understanding self-predictive learning for reinforcement learning." International Conference on Machine Learning. PMLR, 2023.
>
> [2] Khetarpal, Khimya, et al. "A Unifying Framework for Action-Conditional Self-Predictive Reinforcement Learning." International Conference on Artificial Intelligence and Statistics. PMLR, 2025.
>
> [3] Grill, Jean-Bastien, et al. "Bootstrap your own latent-a new approach to self-supervised learning." Advances in neural information processing systems 33 (2020): 21271-21284.
>
> [4] Ni, Tianwei, et al. "Bridging state and history representations: Understanding self-predictive rl." arXiv preprint arXiv:2401.08898 (2024).

---

> > ### Comment · Reviewer_JKTQ · 2025-11-23
> >
> > Thank you for the response; it does mostly address my concerns. I have two remaining points:
> > 1. The additional details about collapse prevention would be very useful to add to the main paper (probably just a few sentences would suffice). In particular, when reading the paper, it was not initially obvious what the relationship is between all of the equations 4-8, and upon re-reading, it does seem that some more explicit description of how BC and BYOL(-$\gamma$) are combined to ensure that "the BC loss itself acts as a strong deterrence against representation collapse" would be helpful. Specifically, the BC loss in (4) does not include the encoder $\phi$ or predictor $\psi$; while (7) does, it is unclear how this shows up in the final loss formula. In other words: where exactly does the impact of $\phi$ and $\psi$ **on the policy** show up in the loss function? Or, is there some other mechanism that "binds" the policy $\theta$ with the encoder and predictor?
> > 2. The response/revision include correlation information, but no plots showing the actual data for the representation similarity's relationship to state distance. Presumably, given that summary statistics were calculated, this information is available. Why not include these plots? (especially given that, a priori, it seems there is no clear reason to assume a linear relationship).
> >     - The new observation that "the ranking of methods in terms of correlation in representation space matches the same ordering of methods in terms of empirical policy performance from Table 2" is quite interesting; if possible, this should be moved to the main text, as it seems like useful information for future work.

---

> ### Author Response · Authors · 2025-11-27
>
> We appreciate your additional questions and would be happy to share some more information.
>
> Follow up to __Q3__: additional details about collapse prevention, “where exactly does the impact of $\phi$ and $\psi$  on the policy show up in the loss function? Or, is there some other mechanism that "binds" the policy  with the encoder and predictor?”
>
> __A3__ continued:
>
> Section 4.2 (Training a policy with auxiliary representations) discusses how the policy utilizes the representations, however we have added more details to the main paper for clarity. Briefly, we have a policy with structure:
>
> $\pi_{\Theta}(a | s,g ) = \text{MLP}_\theta(\text{concat}(\phi(s), \phi(g)))$
>
> For BYOL-gamma, the policy uses the encoder $\phi(\cdot)$ to process state and goal, and then differentiates through both the $\text{MLP}_\theta$ and $\theta$. Thus, the policy directly operates over the representations. Thus, in equation (11),  the term $L \text{-BC}$ updates the parameters of both the policy head $\theta$ and its inputs, i.e. the encoder $\phi$, while $L \text{-aux}$ updates $\psi,\phi$ but not $\theta$. With $\phi$ affected by both terms, the BC loss can prevent collapse of $\phi$ which can occur in BYOL objectives, while the auxiliary loss can prevent overfitting and learn better representations for $\phi$ which help generalization for the policy.
>
>
> Follow up to __Q2__:
>
> __A2__ continued:
> Thank you for this suggestion, we have added the correlation plots to Appendix H.2. This is similar to our computation for the previous table, however we plot fewer points for clarity.
>
> We have also added a reference in the main paper in Section 5.2 to the observation regarding the relationship between empirical success rate. Additionally, in regards to table 12 for correlation we make our claim more precise, in that we list the average success rate of the exact checkpoints used to compute the correlation (which are 4 checkpoints, a subset of those used in Table 2):
>
> | Dataset                    | BYOL-γ      | BYOL        | TRA         | TDSR        |
> |-|-|-|-|-|
> | average maze correlation | 0.66 | 0.27 | 0.32 | 0.53 |
> | average maze success     | 39   | 26   | 31   | 36   |
>
>
>
> We hope these updates increase your assessment of our paper; otherwise, we would be happy to answer any additional questions or concerns!

---

### Official Review · Reviewer_o9Vq · 2025-11-01

**Soundness:** 2
**Presentation:** 3
**Contribution:** 2
**Rating:** 4
**Confidence:** 4

**Summary:**

The authors focus on Goal-Conditioned Behavior Cloning and its inability to “stitch” or as they refer, perform combinatorial generalization. They attribute this inability to the lack of temporal consistency in state representations. The authors look into dynamics aware state representations and connections to successor measures to instill representations that are close for the states visited by can be sampled by successor measure of the policy collecting the data. In other words, the representations of all the states collected by a policy should be close. To ensure scalability, the authors look into BYOL style contrastive methods rather than TD based methods to learn representations that can predict “future” state representations from the policy. The authors further create a few variants of their method using a bidirectional loss. The policy is trained using BC with this BYOL-$\gamma$ being an auxiliary loss.

**Strengths:**

(1) The authors formalize the combinatorial generalization gap or the stitching error using a mixture policy. This formulation is interesting and can be used to study several algorithms and future works.

(2) The authors expand on self-predictive representations from single step or n-step to a general geometric distribution over long horizons. The authors also link this representation to successor measures and show how these representations can be used to estimate successor measures (and features).

**Weaknesses:**

(1) Empirically there is a minor improvement over TD-SR. In visual setting, the performance is similar to GCBC. First of all, how is GCBC stitching? Wasn’t the argument that GCBC cant stitch?

(2) To goal to enforce stitching in GCBC was to come up with an offline RL method that can scale (as the authors claim the TD methods dont). But in visual environments, their method does not scale well too (similar performance to GCBC)

(3) $\psi$ is a network and $\psi^\pi$ is successor features which is confusing, a different symbol should be used.

(4) While the motivation for adding stitching ability to GCBC makes some sense, the motivation for the representation learning objective is not strong. The representation sure can be used to estimate successor measures but does that make them better for BC or does that allow stitching? Are these representations helping with stitching in a theoretical way or its an empirical hypothesis/observation?

(5) Does adding a bidirectional prediction affect the theoretical properties? As the loss changed so the solution would change too.

**Questions:**

(1) The authors say (on page 5, line 238-243) that they want an invariance $\phi(s_f) \approx \phi(s_w)$. This can be fine for the BC policy that was conditioned for $\phi(s_f)$ but would feel is going towards $\phi(s_w)$. But what about the BC policy from $\phi(s_w)$ to $\phi(s_f)$? Wont it assume that it's already at the goal?

(2) How is a collapse avoided? It looks like all states of a trajectory would be mapped close to each other so for the same example, $\phi(s_0) \approx \phi(s_w)$ as they belong to some policy and $\phi(s_f) \approx \phi(s_w)$ which would also imply, $\phi(s_f) \approx \phi(s_0)$. How is a collapse prevented?

(3) Suppose there are four trajectories, $s_0$ to $s_w$, $s_w$ to $s_f$, $s_0$ to $s_y$ and $s_y$ to $s_f$. The trajectory $s_0 - s_y - s_f$ is optimal. Would this method be able to find out the optimal as there is no notion of reward/cost/value?

(4) Are the results statistically significant?

---

> ### Author Response · Authors · 2025-11-22
>
> Thank you for your thoughtful review of our paper. We are glad that you found our method BYOL-$\gamma$ and its connections to the successor measure to be interesting. We will address each of your concerns in turn below.
>
> __Q1 & 2: “Minor improvement over TD-SR, and GCBC in visual environments”.__
>
> A1: First, we have added averages to Table 2, to make it more clear which methods perform better on visual and non-visual environments as we discuss in the meta comment. We see that on non-visual environments, BYOL-gamma (33) and TD-SR (32), make similar improvements over BC (21), while there is a bigger gap on visual environments between BYOL-$\gamma$ (37) and TD-SR (31), which is underperforming BC (34). Thus, we see that BYOL-$\gamma$ performs more consistently across both visual/non-visual tasks.
>
> We also would like to clarify for the reviewer that the contribution of TD-SR used in this setting as an auxiliary loss is novel to our paper. This objective was previously used in work as a pretraining objective, for the zero-shot RL framework [1]. However, it has not been used for representation learning for behavioral cloning objectives, where we make a new observation related to its promise towards combinatorial generalization in Section 4.
>
> __Q3: “network $\psi$ and $\psi^{\pi}_{\phi}$ is successor features which is confusing, a different symbol should be used.”__
>
> A3: For BYOL-$\gamma$, we use this notation because the output of our network $\psi$, has input from basis features $\phi$, relates to approximating the the true successor features with basis features $\phi$, i.e. $\psi^{\pi}_{\phi}$ which we discuss in Section 4.1.
>
> __Q4: “representations helping with stitching in a theoretical way or its an empirical hypothesis/observation?”__
>
> A3: In Section 4., we provide a theoretical intuition for why learning representations with temporal consistency under the successor measure would be beneficial for stitching. In short, we wish to stitch trajectories from $s$ to $s_f$ through a waypoint $s_w$. By encoding $\phi(s_w)$ and $\phi(s_f)$ or more generally $\phi(s_f)$ where $s_f \sim M(s_w, s_f)$ close in representation space, we can encourage this stitching property purely through representations alone.
>
> __Q5: “Does adding a bidirectional prediction affect the theoretical properties?__
>
> A4: The bidirectional prediction objective [4] was proposed to help improve the stability of BYOL, and actually uses a weaker assumption that $(P^\pi)^T$ is a valid transition matrix rather than fully symmetric which we apply to our setting.In [4], the authors show that with bidirectional prediction, the BYOL objective produces representations equivalent to the SVD of the transition matrix. Our results follow on this work, though we only extend their results in the symmetric one-directional case, there is good theoretical motivation that this could likely be extended to the bidirectional case as well.
>
>
> __Q6: Representation Invariance and collapse__
>
> A6: In general, representation collapse can be an issue for self-predictive methods like BYOL [5]. The general strategy for avoiding this collapse is to use a stop gradient and employ an exponential moving average on the target, which has been successful at preventing collapse in other settings, including RL [6][4]. Additionally, we note that for our method, which uses self-predictive representations as an auxiliary loss for BC, the BC loss itself acts as a strong deterrence against representation collapse. The representations passed to the policy are not frozen, rather the  BC loss is allowed to back-propagate through the representations. This encourages representations which are meaningful for predicting expert actions.
>
> [1] Touati, Ahmed, Jérémy Rapin, and Yann Ollivier. "Does Zero-Shot Reinforcement Learning Exist?." The Eleventh International Conference on Learning Representations.
>
> [4] Tang, Yunhao, et al. "Understanding self-predictive learning for reinforcement learning." International Conference on Machine Learning. PMLR, 2023.
>
> [5] Grill, Jean-Bastien, et al. "Bootstrap your own latent-a new approach to self-supervised learning." Advances in neural information processing systems 33 (2020): 21271-21284.
>
> [6] Ni, Tianwei, et al. "Bridging state and history representations: Understanding self-predictive rl." arXiv preprint arXiv:2401.08898 (2024).
>
> ---- continued in next comment ----

---

> > ### Author Response · Authors · 2025-11-22
> >
> > __Q7: Would this method be able to find out the optimal as there is no notion of reward/cost/value?__
> >
> > A7: This is a good example for illustrating our setup. In general, our method is a BC method, where the goal is not to find an optimal policy amongst the existing transitions (as would be the case in offline RL).  Instead, our claim is that vanilla BC would actually not be likely to perform either path from $s_0$ to $s_f$. While we do not claim that our method will induce an optimal path from $s_0$ to $s_f$, it should help the policy pick up one of the two viable paths through representation learning. We induce this behavior in a soft way, in that instead of optimizing the policy towards out-of-distribution goals through maximizing a critic, we build  representation spaces under the policy that reflect distance towards reaching these goals.
> >
> > We provide evidence for this claim through the correlation of a representation space with groundtruth shortest path in maze environments as suggested by reviewer JKTQ which we include in Appendix H.2 and the general response. We can see that the representation space of BYOL-$\gamma$ best aligns with the shortest path in the environment.
> >
> > __Q8:__ Are the results statistically significant?
> >
> > A8: Following previous work on OGBench [2,3], we provide standard deviation across 10 seeds for results in each environment in Table 2. We note that we also bold results within 95% of the value of the best performing method to indicate overlap.
> >
> > Thank you again for your review, and look forward to discussing further.
> >
> > [2] Park, Seohong, et al. "OGBench: Benchmarking Offline Goal-Conditioned RL." The Thirteenth International Conference on Learning Representations.
> >
> > [3] Myers, Vivek, et al. "Temporal Representation Alignment: Successor Features Enable Emergent Compositionality in Robot Instruction Following." arXiv preprint arXiv:2502.05454 (2025).

---

### Official Review · Reviewer_6mmR · 2025-11-03

**Soundness:** 3
**Presentation:** 2
**Contribution:** 3
**Rating:** 4
**Confidence:** 3

**Summary:**

This paper addresses the failure of Goal-Conditioned Behavioral Cloning (GCBC) on tasks requiring combinatorial generalization ("stitching"). The authors propose BYOL-γ, a self-predictive auxiliary loss. The core idea is that this loss encourages the learning of representations that approximate the Successor Representation (SR) without the instability of TD-learning or the pessimism of contrastive learning.

**Strengths:**

- simple, intuitive design.
- well-suited for offline setting (as shown in TD-SR < BYOL-γ).
- good empirical performance.

**Weaknesses:**

- The authors explicitly benchmark only against non-hierarchical methods. However, the current state-of-the-art on the OGBench 'stitch' tasks (e.g., HIQL) is hierarchical. By omitting this comparison, the paper fails to demonstrate how its performance stacks up against the actual SOTA, making its practical significance unclear. It also remains unclear whether BYOL-γ's benefits could be combined with hierarchical methods.


- The empirical improvements on visual-based tasks (visual-antmaze, visual-scene-play) are marginal at best (Table 2). In some cases (e.g., visual-antmaze-large), the proposed BYOL-γ (26.0) performs worse than the standard GCBC baseline (29.2), questioning the method's applicability and benefits for high-dimensional visual inputs.

**Questions:**

None

---

> ### Author Response · Authors · 2025-11-22
>
> Thank you for your thoughtful review of our paper. We are glad that you found our method simple and intuitive. We will address each of your concerns in turn below.
>
> __Q1: “However, the current state-of-the-art on the OGBench 'stitch' tasks (e.g., HIQL) is hierarchical”__
>
> A1: It is true that offline RL methods achieve SOTA, but we argue that contributions that focus on new directions are valuable to the community.  We focus our work on the problem of representation learning to enable stitching using “flat” or non-hierarchical methods.  We believe that this choice is justified since the focus of our paper is not necessarily in achieving SOTA results but in exploring beneficial representation learning losses for GCBC, which can most clearly be understood in the non-hierarchical setting. We also make this choice to be consistent with prior work in this area [1].
> However, we find your suggestion of incorporating representation learning methods like BYOL-$\gamma$ into hierarchical GCBC to be very interesting and we have conducted additional experiments in this setting which we have added to the Appendix C, and highlight here:
>
> | Dataset                    | GCBC  | HGCBC | BYOL-γ | HBYOL-γ |
> |----------------------------|-------|-------|---------|----------|
> | antmaze-medium-stitch      | 45±11 | 60±4  | 61±6    | 76±12    |
> | antmaze-large-stitch       | 3±3   | 11±8  | 21±5    | 29±9     |
> | humanoidmaze-medium-stitch | 29±5  | 35±4  | 54±5    | 61±2     |
> | humanoidmaze-large-stitch  | 6±3   | 4±0   | 14±2    | 21±3     |
> | average                    |    21 |    28 |      38 |       47 |
>
> We can see that hierarchical BYOL-$\gamma$ outperforms standard BYOL-$\gamma$ and standard HGCBC. We include these results, as well as comparison with HIQL, in Appendix C.
>
> __Q2: Performance in visual environments__
>
> A2: Regarding your concerns, we refer you to our overall response to the reviewers for our additional experiments on the visual environments. However, to address your particular concern about worse performance than GCBC in visual-antmaze-large, we note that BYOL-$\gamma$ (26%)  _does_ outperform GCBC (24%) in this environment - the 29% performance you are referring to is for TD-SR, which is also a contribution of our work (as applied in the auxiliary loss setting).
>
> Thank you again for your review, and look forward to discussing further.
>
> [1] Myers, Vivek, et al. "Temporal Representation Alignment: Successor Features Enable Emergent Compositionality in Robot Instruction Following." arXiv preprint arXiv:2502.05454 (2025).

---

> > ### Comment · Reviewer_6mmR · 2025-11-24
> >
> > 1. Thanks for adding the hierarchical results. Could you briefly clarify how HGCBC and HBYOL-γ are implemented?
> > - What exactly is the high-level policy’s input/output, and is BYOL-γ applied only to the low-level or also the high-level?
> > - How frequently are subgoals issued?
> >
> > Also, the hierarchical gains are quite large. Could you explain on why BYOL-γ helps so much in the hierarchical setting (e.g., better subgoal representation, more stable low-level controller, etc.)?
> >
> > 2. For visual tasks, the improvement over GCBC seems small (34 → 37),  so I remain uncertain about how much benefit BYOL-γ actually provides for high-dimensional visual inputs.

---

> > > ### Author Response · Authors · 2025-11-27
> > >
> > > We appreciate your continued feedback and would be happy to answer these questions.
> > >
> > > __Q3__: how HGCBC and HBYOL-γ are implemented?
> > >
> > > __A3__: We have added additional details to the hierarchical setup in appendix C, including a diagram (now Figure 5).
> > > Overall, we build on the HGCBC from [2]. In our setup, the low-level policy is a standard BYOL-γ policy, in fact, we simply load pretrained checkpoints saved from Table 2, giving $\pi^l(a | \phi(s), \phi(l))$. To learn the high-level policy, we freeze the encoder from $\pi_l$, which we label as $\bar\phi$ and train a high-level policy with the fixed representations of the low-level policy: $\pi^h(\bar\phi(l) | \bar\phi(s), \bar\phi(g))$. This policy both uses $\bar\phi$ as an input space for states/goals, and as an output space to predict sub-goals.
> > >
> > > During evaluation, subgoals are re-issued at each timestep following the implementation of HGCBC from [2]. We note that for  HBYOL-γ, this has similar efficiency to our standard BYOL-γ, even in visual environments, as we  can re-use the encoded state representation for both the low-level and high-level policies.
> > >
> > >
> > > __Q4__: visual tasks
> > >
> > > We have performed additional experiments testing HBYOL-γ on visual environments. We have also updated appendix C with these results. Prior work [2] does not perform HGCBC on visual environments, as this entails predicting sub-goals in pixel space. Thus, HBYOL-γ is a natural choice to extend hierarchical policies to pixels.  For another BC baseline on visual, we utilize a similar procedure as HBYOL-γ, but without representation learning (alignment=0), denoted HGCBC-$\phi$.
> > >
> > > We can see that  HBYOL-γ leads to significant improvement over BC on visual environments, and is competitive with work such as HIQL on visual maze environments. We show results below as with more details in  Appendix C:
> > >
> > > | Dataset                      | GCBC    | HGCBC   | HGCBC-$\phi$ | BYOL-$\gamma$ | H-BYOL-$\gamma$ |
> > > |------------------------------|---------|---------|--------------|---------------|-----------------|
> > > | antmaze-medium-stitch        | 45 ± 11 | 60 ± 4  | $\cdot$      | 61 ± 6        | 76 ± 12         |
> > > | antmaze-large-stitch         | 3 ± 3   | 11 ± 8  | $\cdot$      | 21 ± 5        | 29 ± 9          |
> > > | humanoidmaze-medium-stitch   | 29 ± 5  | 35 ± 4  | $\cdot$      | 54 ± 5        | 61 ± 2          |
> > > | humanoidmaze-large-stitch    | 6 ± 3   | 4 ± 0   | $\cdot$      | 14 ± 2        | 21 ± 3          |
> > > | visual-antmaze-medium-stitch | 67 ± 4  | $\cdot$ | 74 ± 6       | 68 ± 4        | 84 ± 8          |
> > > | visual-antmaze-large-stitch  | 24 ± 3  | $\cdot$ | 19 ± 1       | 26 ± 5        | 31 ± 3          |
> > > | visual-scene                 | 12 ± 2  | $\cdot$ | 8 ± 3        | 17 ± 1        | 14 ± 2          |
> > > |||||||
> > > | average-nonvisual            | 21      | 28      | $\cdot$      | 38            | 47              |
> > > | average-visual               | 34      | $\cdot$ | 34           | 37            | 43              |
> > > | average                      | 27      | $\cdot$ | $\cdot$      | 37            | 45              |
> > >
> > > We believe that this experiment shows representations learned by BYOL-$\gamma$ are useful for visual tasks. Our hierarchical setup adds a high-level policy, which shares the same (frozen) representations used by our original setup. Thus, a takeaway may be that our original representation objective is useful for generalization, but a flat policy trained with BC does not fully take advantage of our representations in certain environments.
> > >
> > > We hope our responses have increased your assessment of our paper; otherwise, we would be happy to answer any additional points.

---

### Author Response · Authors · 2025-11-22

We would like to sincerely thank all reviewers for their thoughtful reviews of our paper and insightful comments. In our work, we explore the use of successor measure (SM) learning as an auxiliary loss for GCBC by (1) demonstrating an “intuitive and theoretical” (JKTQ) motivation for SM learning (2) unifying existing approaches TRA and TD-SR under this theory,  the latter of which has not been explored in the GCBC setting, by “showing dynamics aware state representations connections to SM” (o9Vq01) and, (3) most importantly, introducing a novel method BYOL-$\gamma$ which “achieves competitive empirical performance across a suite of challenging tasks from the OGBench benchmarks.” (c69G).

We sincerely appreciate the feedback from reviewers and have incorporated several improvements to our paper which we highlight here:

__Performance in visual environments vs GCBC__

First we have added additional rows to Table 2 to report separate averages for visual and non-visual environments to better illustrate relative performance. Here we can observe that overall in visual environments our method improves over all other auxiliary loss methods, and is the _only_ method to improve over GCBC.

||BYOL-γ | BYOL | TR | TD-SR| GCBC|
|-|-|-|-|-|-|
|Non-visual| 33 | 23 | 26 | 32 |	21|
|Visual| 37 | 32 | 28 | 31 | 34|

We would also like to note that there is an architectural discrepancy between GCBC and BYOL-$\gamma$, namely GCBC employs a joint embedding for state and goal, i,e, $\phi(s, g)$, while our auxiliary loss methods learn separate embeddings $\phi(s)$ and $\phi(g)$. We kept the joint encoding architecture for GCBC as our baseline since this is the original, best performing implementation. However, to more clearly illustrate the benefit of auxiliary loss learning, we conduct experiments on GCBC with split encoders to align directly with BYOL-$\gamma$. We present these results in the table below.

|Dataset|GCBC|GCBC-$\phi$|
|-|-|-|
|antmaze-medium-stitch|45±11|33±5|
|antmaze-large-stitch|3±3|5±4|
|humanoidmaze-medium-stitch|29±5|32±6|
|humanoidmaze-large-stitch|6±3|4±3|
|visual-antmaze-medium-stitch|67±4|37±6|
|visual-antmaze-l|24±3|4±3|
|visual-scene-play|12±2|10±1|
|averagenon-visual|21|19|
|averagevisaul|34|17|
|average|27|18|

Here, we can see more clearly the significant improvement from training with BYOL-$\gamma$.

Though a split architecture is not necessary for GCBC, we note that allowing such an architecture has significant practical benefits, for example in implementing hierarchical methods [1], as suggested by reviewer 6mmR02. Having a separate goal and state encoder for the low-level policy permits us to use the representation $\phi(s)$ as actions in a high-level policy. We experimented with such a set up (HBYOL-$\gamma$ ) by training a low-level policy with our standard BYOL–$\gamma$ setup and a high-level policy $\pi^h$, which takes state and goal representations and predicts an intermediate goal representation $\phi(l)$ (see Appendix C).

|Dataset|GCBC|HGCBC|HGCBC-$\phi$|BYOL-$\gamma$|H-BYOL-$\gamma$|
|-|-|-|-|-|-|
|antmaze-medium-stitch|45±11|60±4|$\cdot$|61±6|76±12|
|antmaze-large-stitch|3±3|11±8|$\cdot$|21±5|29±9|
|humanoidmaze-medium-stitch|29±5|35±4|$\cdot$|54±5|61±2|
|humanoidmaze-large-stitch|6±3|4±0|$\cdot$|14±2|21±3|
|visual-antmaze-medium-stitch|67±4|$\cdot$|74±6|68±4|84±8|
|visual-antmaze-large-stitch|24±3|$\cdot$|19±1|26±5|31±3|
|visual-scene|12±2|$\cdot$|8±3|17±1|14±2|
|||||||
|average-nonvisual|21|28|$\cdot$|38|47|
|average-visual|34|$\cdot$|34|37|43|
|average|27|$\cdot$|$\cdot$|37|45|

We can see that HBYOL-$\gamma$ outperforms other BC setups, specifically non-hierarchical BYOL-$\gamma$ and standard HGCBC.

__Representation quality__

In addition to the qualitative results we provide in Fig 2, per reviewer JKTQ’s suggestion, we have computed the correlation between the distances between states in  representation space and the true shortest path distances (see Appendix H). We can see that on average,  BYOL-$\gamma$’s representations best capture shortest path distance. These correlations also correspond with empirical success rate.

| Dataset                    | BYOL-γ      | BYOL        | TRA         | TDSR        |
|-|-|-|-|-|
| antmaze-medium-stitch         | 0.71 ± 0.01 | 0.59 ± 0.05 | 0.49 ± 0.05 | 0.72 ± 0.03 |
| antmaze-large-stitch| 0.66 ± 0.02 | 0.10 ± 0.02 | 0.62 ± 0.03 | 0.67 ± 0.02 |
| humanoidmaze-medium-stitch    | 0.64 ± 0.02 | 0.18 ± 0.04 | 0.02 ± 0.02 | 0.36 ± 0.03 |
| humanoidmaze-large-stitch| 0.62 ± 0.03 | 0.20 ± 0.03 | 0.15 ± 0.04 | 0.38 ± 0.03 |
| average                    | 0.66        | 0.27        | 0.32        | 0.53        |

We thank all the reviewers again and look forward to further discussion.

[1] Frans, Kevin, et al. "Diffusion Guidance Is a Controllable Policy Improvement Operator." arXiv preprint arXiv:2505.23458 (2025).

---

> ### Author Response · Authors · 2025-11-27
>
> __NOTE TO AC: We have updated the table in our above general response with the table below to facilitate AC review, so that all relevant results are in the above comment. We leave this comment here as a record of our interactions.__
>
> In our discussion with __6mmR__ regarding hierarchical setup, we have performed additional experiments on visual environments. Here, we can see that BYOL-$\gamma$ in a hierarchical setup leads to significant improvement over BC, even in visual maze settings.  We wanted to reply to our general response to reflect the new table.
>
> | Dataset                      | GCBC    | HGCBC   | HGCBC-$\phi$ | BYOL-$\gamma$ | H-BYOL-$\gamma$ |
> |------------------------------|---------|---------|--------------|---------------|-----------------|
> | antmaze-medium-stitch        | 45 ± 11 | 60 ± 4  | $\cdot$      | 61 ± 6        | 76 ± 12         |
> | antmaze-large-stitch         | 3 ± 3   | 11 ± 8  | $\cdot$      | 21 ± 5        | 29 ± 9          |
> | humanoidmaze-medium-stitch   | 29 ± 5  | 35 ± 4  | $\cdot$      | 54 ± 5        | 61 ± 2          |
> | humanoidmaze-large-stitch    | 6 ± 3   | 4 ± 0   | $\cdot$      | 14 ± 2        | 21 ± 3          |
> | visual-antmaze-medium-stitch | 67 ± 4  | $\cdot$ | 74 ± 6       | 68 ± 4        | 84 ± 8          |
> | visual-antmaze-large-stitch  | 24 ± 3  | $\cdot$ | 19 ± 1       | 26 ± 5        | 31 ± 3          |
> | visual-scene                 | 12 ± 2  | $\cdot$ | 8 ± 3        | 17 ± 1        | 14 ± 2          |
> |||||||
> | average-nonvisual            | 21      | 28      | $\cdot$      | 38            | 47              |
> | average-visual               | 34      | $\cdot$ | 34           | 37            | 43              |
> | average                      | 27      | $\cdot$ | $\cdot$      | 37            | 45              |

---

### Meta-Review · Area_Chair_TepZ · 2026-01-06

**Summary:**

This paper addresses the well-known failure of goal-conditioned behavioral cloning (GCBC) to achieve combinatorial generalization (“stitching”) by identifying a clear technical gap: standard BC lacks any inductive bias to encode long-range temporal structure across trajectories. The key technical contribution is a principled auxiliary representation learning objective, BYOL-γ, which injects successor-measure–like temporal consistency into GCBC without relying on TD bootstrapping or contrastive negatives. The paper makes a meaningful conceptual advance by unifying prior auxiliary losses (contrastive learning, TD-SR, standard BYOL) under a successor-measure lens and by showing both theoretically in finite MDPs and empirically on challenging OGBench tasks that self-predictive representations over geometrically sampled future states can approximate long-horizon reachability and thereby enable stitching. Extensive experiments, ablations, and diagnostics (including correlation between representation geometry and shortest-path distance) support the claim that improved representation geometry, rather than policy optimization changes, is the mechanism driving generalization. While some theoretical assumptions are idealized and gains in purely visual flat settings are moderate, the technical insight that successor-aware representation geometry alone can unlock combinatorial generalization in supervised BC is novel and impactful. I therefore recommend acceptance.

**Reviewer Concerns:**

Most concerns were addressed

**Reviewer Scores:**

most reviewer would raise their score.

---

### Decision · Program_Chairs · 2026-01-26

Accept (Poster)